# Double-CRISPR Knockout Simulation (DKOsim): A Monte-Carlo randomization system to model cell growth behavior and infer the optimal library design for growth-based double knockout screens

Yue Gu[1,2], Traver Hart[3], Luis Leon-Novelo[2]*, John Paul Shen[1]*

**1** Department of Gastrointestinal Medical Oncology, The University of Texas MD Anderson Cancer Center, Houston, Texas, United States of America, **2** Department of Biostatistics and Data Science, School of Public Health, University of Texas Health Science Center at Houston, Houston, Texas, United States of America, **3** Department of Systems Biology, The University of Texas MD Anderson Cancer Center, Houston, Texas, United States of America

* Jshen8@mdanderson.org (JPS); Luis.G.LeonNovelo@uth.tmc.edu (LLN)

## Abstract

Advances in functional genomic technology, notably CRISPR using Cas9 or Cas12, now allow for large-scale double perturbation screens in which pairs of genes are inactivated, allowing for the experimental detection of genetic interactions (GIs). However, as it is not possible to validate GIs in high-throughput, there is no gold standard dataset where true interactions are known. Hence, we constructed a Double-CRISPR Knockout Simulation (DKOsim), which allows users to reproducibly generate synthetic simulation data where the single gene fitness effect of each gene and the interaction of each gene pair can be specified by the investigator. We adapted Monte-Carlo randomization methods to extend single knockout simulation methods to double knockout designs, which simulate the gene-gene interactions between all possible combinations of the input genes. Using DKOsim, we generated simulated datasets that closely resemble real double knockout CRISPR datasets in terms of Log Fold Change (LFC), GI distribution, and replicate correlation. We further inferred optimal CRISPR library designs by systematically investigating critical experimental parameters including depth of coverage, guide efficiency, and the variance of initial guide distribution. This simulation scheme will help to identify optimal computational methods for GI detection and aid in the design of future dual knockout CRISPR screens.

## Author summary

We designed DKOsim to simulate CRISPR double knockout screens by modeling cell division behavior with both single knockout (SKO) and double knockout

**Data availability statement:** The authors confirm that all data underlying the findings are fully available without restriction. Codes and the corresponding R package is available on Github (https://github.com/yuegu-phd/DKOsimR). Tutorial on using the package is available at https://dkosimr-tutorial.readthedocs.io/en/v1.0.0/. All relevant data are publicly available within the paper and its Supporting Information files.

**Funding:** This work was supported by the Col. Daniel Connelly Memorial Fund, the Andrew Sabin Family Fellowship Award (J.P.S. is an Andrew Sabin Family Foundation Fellow at The University of Texas MD Anderson Cancer Center), the Cancer Prevention & Research Institute of Texas (RR180035 & RP240392 to J.P.S., J.P.S. is a CPRIT Scholar in Cancer Research), the Appendiceal Cancer Pseudomyxoma Peritonei Research Foundation (Catalyst Research Grant to J.P.S.) and a Conquer Cancer Career Development Award (2022CDA-7604125121 to J.P.S.). Any opinions, findings, and conclusions expressed in this material are those of the author(s) and do not necessarily reflect those of the American Society of Clinical Oncology or Conquer Cancer. The funders had no role in study design, data collection and analysis, decision to publish, or preparation of the manuscript.

**Competing interests:** I have read the journal's policy and the authors of this manuscript have the following competing interests: J.P.S. has consulting/stock ownership in Engine Biosciences and NaDeNo Nanoscience.

(DKO) constructs via Monte-Carlo randomization samplers. Running DKOsim at large scale, we identified the asymptotic tuning points that optimize genetic interaction (GI) identification performance by delta-LFC (dLFC) method compared to the simulated truth. We show that DKOsim is tunable to approximate actual dual-CRISPR knockout screening data. Comparing replicate correlation from DKOsim with experimentally generated data, DKOsim can be tuned based on users' desires to reproduce a similar level of randomness to that observed in variety CRISPR screening conditions.

## Introduction

Clustered Regularly Interspaced Short Palindromic Repeats (CRISPR) was first identified in E. coli bacteria in Japan [1,2]. CRISPR knockout can be multiplexed into a high-throughput genetic screening method to systematically perturb genes and/or pairs of genes [3]. In 2017, combinatorial CRISPR-Cas9 screens were performed in cancer cells for the first time to allow for high-throughput identification of synthetic lethal gene pairs [4,5]. More recently, the Cas12a platform provides a highly efficient multiplex gene knockout, significantly increasing efficacy of gene knockout and decreasing library size and thus the cost of genetic interaction screening [6,7]. Combinatorial CRISPR technology has revolutionized the discovery of gene-gene interactions [defined as genetic interactions (GIs)] by allowing large-scale screening in human cell lines, organoids, and mouse models. GIs occur when the fitness effect of a gene is modified by the functional status of other genes. It is measured by comparing fitness following the CRISPR knockout of a gene pair (double knockout, DKO) vs. the knockout of each gene (single knockout, SKO). The study of GIs reveals the functional relationships between genes and pathways, specifically, the synthetic interactions and compensatory pathways. The findings serve as the foundation for systematic gene network construction, which is valuable for novel drug development in cancer research [8].

With around 200 million possible interacting gene pairs for a mammalian cell, GIs are typically rare and hard to quantify accurately from noisy data [8,9]. Hence, there is limited consensus on the use of existing computational tools for detecting GIs. Moreover, experimental validation of GIs in high-throughput is difficult since there are no gold standard datasets that include the true interactions. To address these challenges, we developed a probabilistic simulation framework with simulated theoretical GI truth to emulate the real laboratory CRISPR screening procedures for both SKO and DKO designs. This approach allows us to efficiently test several interacting gene pairs for GIs and approximate the underlying true GI distributions.

Prior efforts in generating simulation frameworks for CRISPR screening mainly used SKO designs. In 2015, Stombaugh et al. proposed the Power Decoder Simulator [10] to generate in silico shRNA pooled screening experiments using short hairpin RNA (shRNA) to efficiently estimate the genotypic biological relevance of a set of genes based on their experimental phenotype. Building upon this, Nagy et al. (2017)

developed the first CRISPR SKO discrete simulation tool, CRISPulator [11], which simulates both the Growth-based and the Fluorescence-Activated Cell Sorting (FACS)-based SKO screening to test the effects of different library designs. de Boer et al. (2020) designed MAUDE [12] (Mean Alterations Using Discrete Expression) utilizing CRISPulator to show the consistency in the optimal cell-sorting bins configurations by quantiles via simulation, and derive the mean expression of cells containing each guide. These studies connected the designed simulation framework with empirical assumptions from real experiments but lacked systematic profiling of the screening parameters for growth-based pooled CRISPR screens that assume exponential cell growth. Moreover, since they did not consider the combinatorial CRISPR screening (i.e., DKO) design, the effects of GIs are not articulated in their simulated CRISPR design.

To address the need for systemizing a CRISPR simulation scheme and the lack of a DKO simulation framework, we developed a Double-CRISPR Knockout Simulation (DKOsim) that will enable researchers to determine the DKO behavior, including the simulated theoretical true interactions. DKOsim will help identify the optimal GI detection methods and design future double-guided CRISPR experiments. The motivations of the study are visualized in Fig 1. We started by analyzing sets of CRISPR DKO screening data using two different computational approaches named CTG [13] and GEMINI [14]. While the SKO gene fitness scores showed high correlation, the GI scores were essentially random, with almost no overlap in the identified GIs between the two methods in HeLa cell line. Without the GI ground truth, we could not determine which method was detecting the truly interacting gene pairs. Motivated by this, we designed a systematic synthetic data simulation scheme to simulate theoretical GIs that could be treated as the underlying truth.

## Methods

### Overview and notations

We simulated the CRISPR knockout screens by modeling cell division, using Monte-Carlo randomization sampling. The common notations and conceptual methodology of the simulation scheme are summarized in Fig 2. We indexed genes with $k \in \{1, \ldots, n\}$ and used the notation $SKO(k)$ to refer to a cell with only gene $k$ to be knocked out. We used the notation

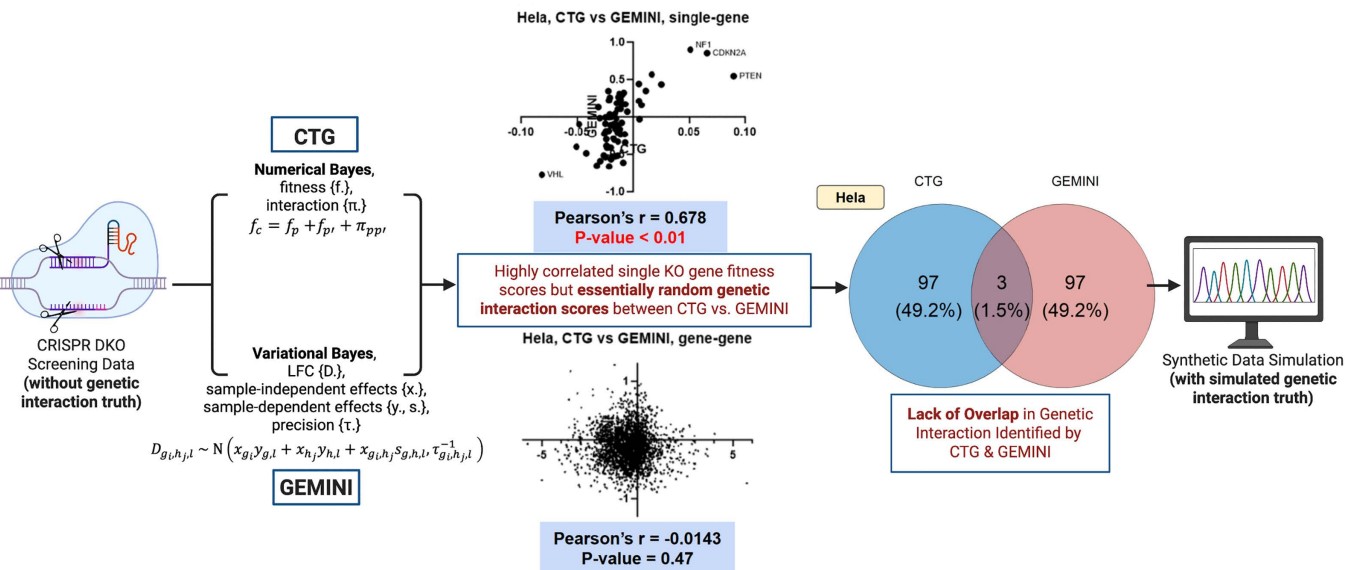

**Fig 1. Motivations of the study.** Two genetic interaction (GI) identification methods named CTG and GEMINI are applied to HeLa cell line in Shen et al. 2017 Double-CRISPR Knockout (DKO) datasets. Computational results are visualized in both scatterplots of CTG vs. GEMINI scores, and Venn diagram on the identified GI by CTG and GEMINI. Created in BioRender. Gu, **Y.** (2026) https://BioRender.com/elgwcy7.

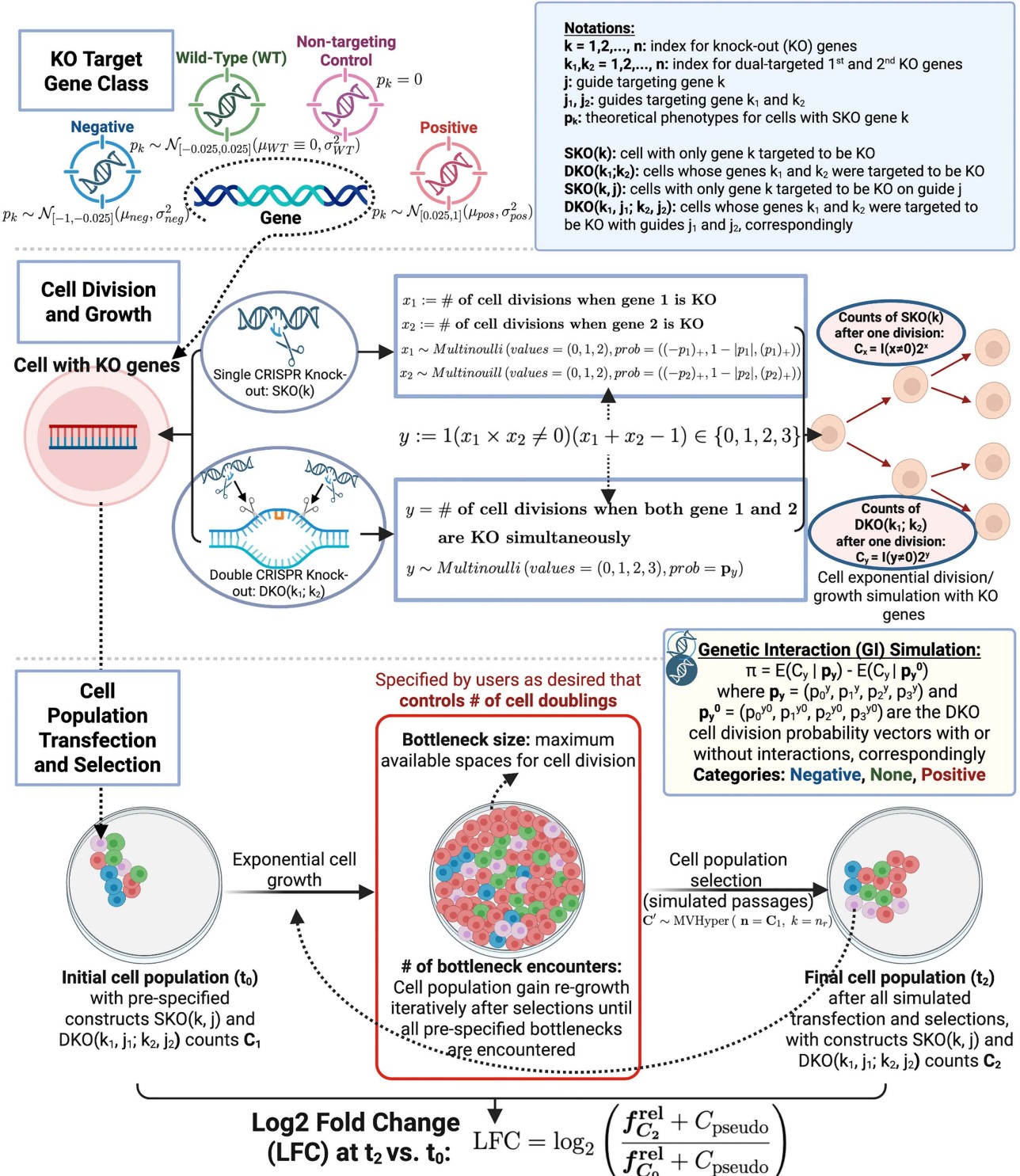

**Fig 2. Simulation methodology.** Main modules are conceptually visualized in the simulation schematic design, including KO target gene class initialization, cell division and growth modelling, and cell population transfection and selection simulation. Created in BioRender. Gu, **Y.** (2026) https://BioRender.com/tl5sg1s.

$DKO\left(k_1; k_2\right)$ to refer to cells whose genes $k_1$ and $k_2$ were targeted for knock out. Since the order of knocking out the genes does not matter, we used $k_1$ and $k_2$, with $k_1 < k_2$, $k_1, k_2 \in \{1, \ldots, n\}$ to index the dual-targeted genes in a DKO cell.

We extended the notation to consider the guides targeting specific genes to be knocked out. We referred to the definition of guide in CRISPR screening as a synthetic RNA molecule named guide-RNA (gRNA) that directs CRISPR-associated nuclease, such as Cas9 or Cas12, to a target DNA sequence [15]; in SKO cells, we assumed a single-guide (sgRNA) disrupts one gene, whereas in DKO screens, two distinct guides were combined as one dual-guide (dgRNA) to perturb two genes simultaneously. We additionally defined the construct as either gene and guide or the combination of two genes and their corresponding guides. In this way, the guide targeting gene $k$ is indexed with $j$ and the notation $SKO(k, j)$ refer to cells on construct with only gene $k$ targeted with guide $j$. Note that in $SKO(k, j)$ and $SKO\left(k', j\right)$, though the guides share the same index $j$, are different. Similarly, $j_1$ and $j_2$ refer to guides targeting genes $k_1$ and $k_2$ simultaneously. $DKO\left(k_1, j_1; k_2, j_2\right)$ denote cells on constructs whose genes $k_1$ and $k_2$ are targeted to be knocked out with guides $j_1$ and $j_2$, correspondingly.

We initialized the knockout (KO) target gene into one of the four main classes: Negative, Wild-Type (WT), Non-targeting Control, or Positive. The theoretical phenotype of each class of genes was drawn from pre-specified distributions, where the mean and variance of each gene class can be tuned by users as desired. Treating the initialized genes as inputs, we derived the cell division and growth behavior in both $SKO(k)$ and $DKO\left(k_1; k_2\right)$ designs from Multi-bernoulli (Multinoulli) distributions that model the exponential cell growth with KO target genes. To simulate cell population transfection and selection procedure, we additionally incorporated the guide-efficacy effects, and defined the initial cell population at $t_0$ as the population that contains all pre-specified constructs $SKO(k, j)$ and $DKO\left(k_1, j_1; k_2, j_2\right)$ with set of counts $C_1$. After several cell doublings controlled by the users' desires on bottlenecks, the final cell population at $t_2$ contained $SKO(k, j)$ and $DKO\left(k_1, j_1; k_2, j_2\right)$ with set of counts $C_2$. We calculated the Log2 Fold Change (LFC) at $t_2$ vs. $t_0$ to quantify the change in the relative abundance of the constructs over time. We also calculated the simulated true GI $\pi$ by measuring the difference between the expected counts given DKO cell division probability vectors, with or without interactions.

We summarized the analytical framework of the study in Fig 3, based on users' inputs. DKOsim mimics the real CRISPR screening for both SKO and DKO designs that output the simulated growth-based screening data and

**Fig 3. Study design.** Analytical frameworks of the study. Created in BioRender. Gu, **Y.** (2026) https://BioRender.com/fek72q6.

calculates the LFC for all constructs with simulated theoretical GI truth. A standard analytical workflow for the simulated datasets includes SKO genes and LFC distributions visualizations, DKO gene combinations deconvolution and dLFC application. Here, we referred the term gene combination deconvolution to the signal-stratification of aggregate LFC measurements into distributions corresponding to distinct gene-class pairings, thereby isolating overlapping phenotypic signals [16].

**Cell-behaviors multinoulli distribution derivations**

**Growth behavior derivations of SKO cells.** For one unit of the cellular population doubling cycle, for a single cell, we define

$$x_1 = \text{\# of cell divisions when gene 1 is KO}$$

$$x_2 = \text{\# of cell divisions when gene 2 is KO}$$

Specifically, the knockout of gene 1 will yield, in terms of cell division, one of the following three outcomes, in one unit of WT cell doubling time: (a) $x_1 = 0$: Cell does not divide and loses viability; (b) $x_1 = 1$: Cell divides once as WT; (c) $x_1 = 2$: Cell divides twice.

These outcomes are simplified for further derivations and simulation programming. We adapted the ideas from CRISPulator; it is possible that the cell divides more than twice or that the cell starts division, but takes twice as long to divide as a WT cell. We did not choose the option "cell does not divide and remains viable", allowing us to connect our discrete simulation approach with the continuous exponential growth-based model (S1 Text, Connection between the Discrete and the Continuous Model).

Given $x_1 \in \{0, 1, 2\}$, the SKO cell will produce $C_{x_1} = \mathbf{1}(\mathbf{x_1} \neq \mathbf{0}) \times 2^{x_1} \in \{0, 2, 4\}$ descendants. The value of $x_1$ depends on the theoretical phenotype $p_1 \in [-1, 1]$, and we assume

$$x_1 \sim Multinoulli\left(values = (0, 1, 2), prob = \left((-p_1)_+, 1 - |p_1|, (p_1)_+\right)\right) \tag{1}$$

where $(a)_+ := \max\{0, a\}$, and the $Multinoulli(values, prob)$ refers to the multi-bernoulli distribution where $x = value[j]$ with probability $prob[j]$.

Based on the value of $p_1$, there are three possible outcomes: (a) When $p_1 < 0$, it is a negative phenotypic gene with $x_1 \in \{0, 1\}$, where $p(x_1 = 0) = -p_1$, $p(x_1 = 1) = 1 + p_1$. (b) When $p_1 = 0$, the cell behaves like a WT cell with $x_1 \equiv 1$. (c) When $p_1 > 0$, $x_1 \in \{1, 2\}$, it is a positive phenotypic gene where $p(x_1 = 1) = 1 - p_1$, $p(x_1 = 2) = p_1$.

Without loss of generality, the same deductions and notations are used for $x_2$ when we knock out gene 2 as in (1), but with parameter $p_2$ where

$$x_2 \sim Multinouill\left(values = (0, 1, 2), prob = \left((-p_2)_+, 1 - |p_2|, (p_2)_+\right)\right) \tag{2}$$

**Growth behavior derivations of DKO cells.** For DKO gene-level outcomes, we considered the joint effects of both the variables $x_1$ and $x_2$, with parameters $p_1$ and $p_2$, respectively. Similar to the notations for SKO effects, we included a variable $y$ for DKO effects. More specifically, for one unit of the cellular population doubling cycle, for a specific single cell, we denote

$$y = \text{\# of cell divisions when both gene 1 and gene 2 are KO simultaneously}$$

in one unit of the cell doubling time, we assume $y$ will have one of the following outcomes: (a) $y = 0$: Cell does not divide and loses viability; (b) $y = 1$: Cell divides once as wildtype; (c) $y = 2$: Cell divides twice; (d) $y = 3$: Cell divides three times.

As such, given $y \in \{0, 1, 2, 3\}$, under no gene-gene interaction $x_1 \perp x_2$. We define

$$y := 1\left(x_1 \times x_2 \neq 0\right)\left(x_1 + x_2 - 1\right) \in \{0, 1, 2, 3\} \tag{3}$$

The original single cell in the DKO design produces $C_y = 1\left(y \neq 0\right) \times 2^y \in \{0, 2, 4, 8\}$ descendants per WT population doubling cycle. We chose this definition of $y$ so that if the first targeted gene is a non-targeting control (this is, $p_1 \equiv 0$, $x_1 \equiv 1$ and there is no gene-gene interaction), then DKO cells behave like SKO cells based upon the second-targeted-gene, regardless of the behavior of the first targeted gene. In math, $x_2$ follows the multinoulli distribution in (2) and $y \sim x_2$. This is shown below in (8).

The distribution of $y$ is

$$y \sim Multinoulli\left(values = (0, 1, 2, 3), prob = \mathbf{p}_y\right) \tag{4}$$

where $\mathbf{p}_y = \left(p_0^y, p_1^y, p_2^y, p_3^y\right)$ is the cell division probability vector.

***Deriving the values of the cell division probability vector*** $\left(p_0^y, p_1^y, p_2^y, p_3^y\right)$. Based on the definition of additive interactions in combination perturbation [17], we calculated $y$ as shown in (3). Its value as a function of $x_1$ and $x_2$, is given in the following matrix:

| $x_1 \backslash x_2$ | 0 | 1 | 2 |
|---|---|---|---|
| 0 | 0 | 0 | 0 |
| 1 | 0 | 1 | 2 |
| 2 | 0 | 2 | 3 |

Accordingly, the joint density of $(x_1, x_2)$ is given by the matrix

| x1\x2 | 0 | 1 | 2 |
|---|---|---|---|
| 0 | $a_{11}$ | $a_{12}$ | $a_{13}$ |
| 1 | $a_{21}$ | $a_{22}$ | $a_{23}$ |
| 2 | $a_{31}$ | $a_{32}$ | $a_{33}$ |

where $a_{ij} \geq 0$ and $\sum_{i=1}^{3} \sum_{j=1}^{3} a_{ij} = 1$. So $\mathbf{p_y}$ in (4) is

$$p_0^y = a_{11} + a_{12} + a_{13} + a_{21} + a_{31},$$
$$p_1^y = a_{22},$$
$$p_2^y = a_{23} + a_{32},$$
$$p_3^y = a_{33} \tag{5}$$

The $\left(p_0^y, p_1^y, p_2^y, p_3^y\right)$ can take any values in $[0, 1]$ as long as they add up to 1. Under the condition of no gene-gene interactions, where $x_2$ and $x_2$ are independent (in math $x_1 \perp x_2$), the matrix above becomes

| x1\x2 | Prob | 0<br>(−p2)+ | 1<br>1−|p2| | 2<br>(p2)+ |
|---|---|---|---|---|
| 0 | (−p1)+ | (−p1)+(−p2)+ | (−p1)+[1−|p2|] | (−p1)+(p2)+ |
| 1 | 1−|p1| | [1 − |p1|](−p2)+ | [1 − |p1|][1−|p2|] | [1 − |p1|]<br>(p2)+ |
| 2 | (p1)+ | (p1)+(−p2)+ | (p1)+[1−|p2|] | (p1)+(p2)+ |

Thus, under no interaction $x_1$ and $x_2$ induce the multinoulli distribution shown in (4), and for $y$ with $\mathbf{p}_y = \mathbf{p}_y^0$, this is

$$y \sim Multinoulli\left(values = (0, 1, 2, 3), prob = \mathbf{p}_y^0\right) \tag{6}$$

With $p_y^0 := \left(p_0^{y_0}, p_1^{y_0}, p_2^{y_0}, p_3^{y_0}\right)$ where

$$
\begin{aligned}
p_0^{y_0} &= (-p_1)_+ (-p_2)_+ + (-p_1)_+ \left[1 - |p_2|\right] + (-p_1)_+ (p_2)_+ + \left[1 - |p_1|\right] (-p_2)_+ + (p_1)_+ (-p_2)_+, \\
p_1^{y_0} &= \left[1 - |p_1|\right] \left[1 - |p_2|\right], \\
p_2^{y_0} &= \left[1 - |p_1|\right] (p_2)_+ + (p_1)_+ \left[1 - |p_2|\right], \\
p_3^{y_0} &= (p_1)_+ (p_2)_+
\end{aligned}
\tag{7}
$$

So when $p_1 = 0$, and there is no interaction (*i.e.*, when the first targeted gene is a non-targeting control),

$$y \sim Multinoulli\left(values = (0, 1, 2, 3), prob = (p_0^{y_0} = (-p_2)_+, p_1^{y_0} = 1 - |p_2|, p_2^{y_0} = (p_2)_+, p_3^{y_0} = 0)\right) \tag{8}$$

That is the same distribution of $x_2$ in (2). Also note that the multinoulli distribution of $x_1$ in (1) is the same as the multinoulli distribution of $y$ in (6) with $p_2 \equiv 0$. In a later subsection, we describe the simulation of the number of divisions in an SKO or DKO cell without gene interaction from the multinoulli distribution shown in (6). We also used (6) to simulate cell divisions of a DKO cell, with interaction jiggling the values of $p_1$ and $p_2$.

**Genetic interaction derivations.** To simulate the gene-gene interactions, we define the theoretical GI based on the growth rate of the cells [18] as

$$\pi := E\left(C_y \mid \mathbf{p}_y\right) - E\left(C_y \mid \mathbf{p}_y^0\right) \tag{9}$$

where, $C_y$ denotes the number of descendants of the DKO cell, defined as shown in (3), after one unit of population doubling cycle. $\mathbf{p}_y^0$ is defined in (7) and $\mathbf{p}_y = (p_0^y, p_1^y, p_2^y, p_3^y)$ is calculated by $p_i^I$ defined as follows:

Based on the GI flag $I \in \{0, 1\}$ (no interaction coded as $I = 0$ and interaction as $I = 1$) from the initial cell library detailed in Simulation System Design I, we define the resampled theoretical phenotypes with interactions as

$$p_i^I \sim \mathcal{N}_{[-1,1]}(p_i, \sigma_{GI}^2)$$

where $I = 1$, and $i = 1, 2$ indexes the first and second KO genes in the cell. We defined $p_0^y, p_1^y, p_2^y, p_3^y$ as the resulting $p_0^{y_0}, p_1^{y_0}, p_2^{y_0}, p_3^{y_0}$ when replacing $p_1, p_2$ by $p_1^I, p_2^I$ in (7), and the simulated GI $\pi$ as shown in (9). When $I = 0$, we defined $p_i^I := p_i$ for $i = 1, 2$ and then $\pi = 0$.

$$\pi = 0(p_0^y - p_0^{y_0}) + 2(p_1^y - p_1^{y_0}) + 2^2(p_2^y - p_2^{y_0}) + 2^3(p_3^y - p_3^{y_0})$$

### Simulation system design I: Cell library construction

**Parameters specification.** Table 1 summarizes all tunable parameters used as inputs in our CRISPR KO simulation scheme. The glossary of each tunable component and its deduced products in our designed system is detailed in S1 Text, Glossary of tunable components.

**SKO gene initialization.** As defined in (1), we first initialized the theoretical phenotypes for the SKO cells, $p_k \in [-1, 1]$, $k = 1, \ldots, n$. Each *SKO(k)* corresponds to one of four classes: negative, positive, wild-type (WT), and non-targeting

**Table 1. Summary table of tunable parameters in the simulation system[1].**

| Parameters | Components of the System |
|---|---|
| $n$ | Number of unique genes |
| $C$ | Coverages: cell representation per guide |
| $n_g$ | Number of guides per gene |
| %GI | Percentage of GIs |
| $\sigma_{GI}$ | Strength of the simulated GIs |
| Percentage of each gene type[2]: %neg | Percentage of Genes in Negative Class |
| %pos | Percentage of Genes in Positive Class |
| %wt | Percentage of Genes in Wild-Type Class |
| %ctrl | Percentage of Genes in Non-Targeting Control Class |
| Theoretical phenotypes parameters[3]: $\mu_{neg}, \sigma_{neg}$ | Mean and SD of Negative Genes |
| $\mu_{pos}, \sigma_{pos}$ | Mean and SD of Positive Genes |
| $\sigma_{wt}$ | SD of Wild-Type Genes |
| $\sigma_f$ | Dispersion of the initial frequency of SKO counts[4] |
| %heg | Percentage of guides with high efficacy |
| Guide-efficacy parameters[5]: $\mu_{high}, \sigma_{high}$ | Mean and SD of guides with high efficacy |
| $\mu_{low}, \sigma_{low}$ | Mean and SD of guides with low efficacy |
| Mode: CRISPRn/CRISPRn-100%Eff | Mode of CRISPR knockout screens: <br> • CRISPRn: mode incorporating guide-efficacy drawn from random distribution <br> • CRISPRn-100%Eff: mode assuming all guides are 100% efficient without any randomization |
| Cell Doublings Parameters[6]: $n_b$ | Bottleneck Size |
| $n_e$ | Number of Bottleneck Encounters |

Abbreviations: GIs, Genetic Interactions; SD, Standard Deviation; SKO, Single knockout;

1. We assume a maximum of 30 doubling cycles for each simulated experiment with moi $\lambda = 0.3$.

2. The percentages of gene types (%$_{neg}$, %$_{pos}$, %$_{wt}$, %$_{ctrl}$) should sum up to 100%.

3. We are referring to the mean and sd of normal distribution before being truncated to generate the theoretical phenotypes of each class of genes.

4. We control the dispersion of the initial constructs' counts by $\sigma_f$. By default, $\sigma_f = \frac{1}{3.29}$, chosen by setting a 10-fold difference between the 95 and 5 percentiles of the SKO counts distribution.

5. We are referring to the mean and sd of normal distribution before being truncated to generate the guide-efficacy of each guide.

6. Both bottleneck size $n_b$ and encounters $n_e$ control bottleneck-induced selection pressure: we utilize these to control the cell doubling time. We simplify $n_b$ to an integer that indicates how many times it is relative to the initial library size. The total number of cell doublings in each simulation run is tracked by a log file under the running directory.

control. The proportion of $SKO(k)$s of each class is specified by the user. The distribution of $p_k$ within each class is also prespecified by the user. Specifically, we assumed the distribution for phenotypes $p_k$, $k = 1, \ldots, n$ is:

1. Negative: $p_k \sim \mathcal{N}_{[-1,-0.025]}(\mu_{neg}, \sigma_{neg}^2)$

2. Positive: $p_k \sim \mathcal{N}_{[0.025,1]}(\mu_{pos}, \sigma_{pos}^2)$

3. WT: $p_k \sim \mathcal{N}_{[-0.025,0.025]}(\mu_{WT} \equiv 0, \sigma_{WT}^2)$

4. Non-Targeting Control: $p_k = 0$

where $\mathcal{N}_{[a,b]}(\mu, \sigma^2)$ denotes the normal distribution with mean $\mu$ and variance $\sigma^2$ truncated to the interval $[a, b]$.

Second, for each $SKO(k)$, we defined $f_0'^k > 0$ that would later help us determine the initial relative frequency of each (SKO and DKO) construct in the library. We obtained $f_0'^k$ by drawing

$$\log_{10}\left(f_0'^k\right) \overset{\text{iid}}{\sim} N\left(0, \sigma_f^2\right) \quad k = 1, \ldots, n$$

where $\sigma_f$ is the standard deviation of the log10 normal distribution of $f_0'^k$. Following CRISPulator [11], by default, $\sigma_f = \frac{1}{3.29}$ where the number 3.29 was chosen so that there is a 10-fold difference between the 95 and 5 percentiles of the initial SKO counts distribution. As a toy example, shown in S1 Table, we generated a library with $n = 3$ genes, 1,2, and 3 belonging to negative, WT, and non-targeting control gene classes, respectively. After sampling from the above distributions, we obtained $p_1 = -0.4$, $p_2 = -0.03$ and $p_3 = 0$.

**DKO gene initialization.** To initialize the cell library $L_{gene}^0$ containing both SKO and DKO cells, we generated all unique combinations of gene pairs that can be targeted for KO, $DKO(k_1; k_2)$ with $k_1, k_2 \in \{1, 2, \ldots, n\}$, $k_1 < k_2$. Then we row binded the $SKO(k)$s and $DKO(k_1; k_2)$s to generate all indices for genes $k = 1, \ldots, n$ and their unique combination in pairs.

We aimed to generate a set of preliminary (non-standardized) frequencies for SKOs and DKOs. We defined a preliminary non-standardized (*i.e.*, they do not add up to 1) frequency of cells with SKOs and DKOs. At the initial timepoint $t_0$, this frequency is defined as $f_k = f_0'^k$ for $SKO(k)$ and $f_{k_1, k_2} = (f_0'^{k_1} + f_0'^{k_2})/2$ for $DKO(k_1; k_2)$. These frequencies will be used later to generate the initial library counts.

**Genetic interaction index initialization.** For every $DKO(k_1; k_2)$ an interaction indicator $I_{k_1, k_2}$ was generated. Within the set of DKOs not containing a non-targeting control, we randomly selected $\%_{GI}$ of DKOs and flagged their genes as interacting (coded as $I = 1$) and the rest as not interacting ($I = 0$). Recall that $\%_{GI}$, selected by the user, is the proportion of DKO constructs with no non-targeting controls whose genes interact. For each $DKO(k_1; k_2)$ we defined $p_{k_1}^I$ and $p_{k_2}^I$, that later helped us define interaction, as $p_{k_i}^I := p_{k_i}$ for $i = 1, 2$ if there is no interaction, $I_{k_1, k_2} = 0$; and we draw $p_{k_i}^I \sim \mathcal{N}_{[-1,1]}(p_{k_i}, \sigma_{GI}^2)$, with $i = 1, 2$ if there is interaction, $I_{k_1, k_2} = 1$. The user-specified simulation parameter $\sigma_{GI}^2$ will control the magnitude of the GIs in the gene pairs. Continuing with our toy example, let's assume that we are requesting that $\%_{GI} = 100\%$, *i.e.*, 1st and 2nd targeted genes $k_1, k_2$ interact. S2 Table shows an example of 3 unique genes drawn from the above distributions, with all possible combinations without considering orders.

**Guides initialization.** We initialized the guides targeting each gene and categorized them based on their KO efficiency. The efficacy of the guides $j$ targeting gene $k$ was denoted as $Eff_{g_{k,j}}$. We added an index $H$ or $L$, to differentiate between high ($Eff_{g_{k,j}^H}$) and low ($Eff_{g_{k,j}^L}$) efficiency guides. Guide-efficacy (high/low) of $n \times n_g$ (*i.e.*, # of genes$\times$# of guides targeting each gene) guides was determined by randomly selecting $\%_{heg} \times n \times n_g$ guides to be highly efficient and the rest to be low efficient. Recall $\%_{heg}$ is the percentage of high-efficacy guides chosen by the user. Based on the simulated guide-efficacy (high or low), the guide-efficacy was simulated by a tunable CRISPR model parameter, as summarized below:

1. **CRISPRn** (CRISPR-nuclease):

$$Eff_{g_{k,j}^H} \sim \mathcal{N}_{[0.6,1]}(\mu_{high}, \sigma_{high}^2)$$

$$Eff_{g_{k,j}^L} \sim \mathcal{N}_{[0,0.6]}(\mu_{low}, \sigma_{low}^2)$$

There is no experimental evidence supporting the threshold of 0.6. Nevertheless, both mean efficiency of high and low-efficacy guides $\mu_{high}$ and $\mu_{low}$ are tunable by users, with default values 0.9 ($\sigma_{high} = 0.1$), and 0.05 ($\sigma_{low} = 0.07$). The

guide efficacy for both high and low categories drawn from randomization almost never reaches 0.6 by default. This threshold is used to ensure that the overlapping probability is exactly zero, thereby guaranteeing that the high-efficacy guides always have higher efficacy than the low-efficacy guides.

2. **CRISPRn-100%Eff** (CRISPR-nuclease with full-efficacy guides): $Eff_{g_{k,j}} = 1$

***Constructs frequency and counts initialization.*** We utilized the Dirichlet Distributions to randomly assign the initialized cell counts to guides as follows:

1. For SKO, we determine the non-standardized relative frequency $f_{g_{k,j}}$ of cells with guide $j$ targeting gene $k$, *i.e.*, $SKO(k, j)$ for $k = 1, \ldots, n$ and $j = 1, \ldots, n_g$:

$$\left( f_{g_{k1,1}}, f_{g_{k1,2}}, \ldots, f_{g_{k,n_g}} \right) = f_k \times Dirichlet \left( \alpha \times \mathbf{1}_{n_g} \right) \tag{10}$$

s.t. $\sum_{j=1}^{n_g} f_{g_{k,j}} = f_k$

2. For DKO, we computed the non-standardized relative frequency $f_{g_{k_1,j_1;k_2,j_2}}$ of cells with a guide $j_1$ targeting gene $k_1$ and guide $j_2$ targeting gene $k_2$, *i.e.*, $DKO(k_1, j_1; k_2, j_2)$ for $k_1 = 1, \ldots, n-1$, $k_2 = k_1 + 1, \ldots, n$ and $j_1, j_2 = 1, \ldots, n_g$:

$$\left( f_{g_{k_1,j_1;k_2,j_2}} \right) = f_{k_1,k_2} \times n_g \times Dirichlet \left( \alpha \times \mathbf{1}_{n_g \times n_g} \right) \tag{11}$$

s.t.

$$\frac{1}{2n_g} \left( \sum_{j_1=1}^{n_g} f_{g_{k_1,j_1}} + \sum_{j_2=1}^{n_g} f_{g_{k_2,j_2}} \right) = \frac{1}{n_g^2} \sum_{j_1=1}^{n_g} \sum_{j_2=1}^{n_g} f_{g_{k_1,j_1;k_2,j_2}}$$

For each pair of genes, across the guides, the average relative frequency of the SKOs is the same as the average relative frequency of DKOs. Hence, the initial counts of SKO and DKO are similar. Also, the distribution of $SKO(k_1)$ Log2 fold change to be similar to the distribution of $DKO(k_1; k_2)$ when the gene $k_2$ is a non-targeting control (*i.e.*, no interaction).

Specifically, we chose $\alpha = 100$ for generating $Dirichlet(\alpha \times \mathbf{1}_{n_g})$ in (10) and $Dirichlet(\alpha \times \mathbf{1}_{n_g \times n_g})$ in (11) to maintain a small variance in the relative frequencies of the SKO and DKO counts across the guides within each gene or gene-pair, respectively.

Similar to the DKO gene initialization, we calculated the relative frequency of each construct (*i.e.*, combinations of the targeting SKO(DKO) gene(s) and the corresponding guides) at $t_0$ as

$$f_{g_{k,j}}^{\text{rel}} = \frac{f_{g_{k,j}}}{D} \quad \text{for } SKO(k, j) \quad f_{g_{k_1,j_1;k_2,j_2}}^{\text{rel}} = \frac{f_{g_{k_1,j_1;k_2,j_2}}}{D} \quad \text{for } DKO\left( k_1, j_1; k_2, j_2 \right) \tag{12}$$

with $k, k_1, k_2 \in \{1, \ldots, n\}$, $k_1 < k_2$ and $j, j_1, j_2 \in \{1, \ldots, n_g\}$. $D := \sum_{k=1}^{n} \sum_{j=1}^{n_g} f_{g_{k,j}} + \sum_{k_2=2}^{n} \sum_{k_1=1}^{k_2-1} \sum_{j_1=1}^{n_g} \sum_{j_2=1}^{n_g} f_{g_{k_1,j_1;k_2,j_2}}$ so that the relative frequencies sum up to 1.

The initial construct counts are set equal to $c_0^{g_{k,j}} = \lfloor f_{g_{k,j}}^{\text{rel}} \times L_0 \rceil$ for $SKO(k, j)$ and $c_0^{g_{k_1,j_1;k_2,j_2}} = \lfloor f_{g_{k_1,j_1;k_2,j_2}}^{\text{rel}} \times L_0 \rceil$ for $DKO(k_1, j_1; k_2, j_2)$, respectively, where again $\lfloor x \rceil$ denotes the rounding of the real number $x$ to the nearest integer, $n$ is the total number of unique single input genes, $n_g$ is the total number of guides corresponding to each unique KO construct, and $L_0$ is the requested initial library size specified by the user. Our initialized cell library with constructs from all pre-specified guides and genes is denoted as $L_{guide}^0$. S3 Table shows the toy example of the simulated cell library counts with $n_g = 2$ initialization with 18 row based on S2 Table, using a coverage $C = 100$ yielding a library size of $L_0 = 1800$.

**Cell growth behavior of SKO and DKO constructs incorporating guide-efficacy.** Following the simulation of cell behavior with KO genes, incorporating both the guide-efficacy and GI effects, we defined

$$p_{k,j}^{I'} = p_k^I \times Eff_{g_{k,j}} \tag{13}$$

for $SKO(k,j)$, $k = 1, \ldots, n$ and $j = 1, \ldots, n_g$. The cell growth behavior of $SKO(k,j)$ will be determined by $p_{k,j}^{I'}$ while the $DKO(k_1,j_1;k_2,j_2)$ behavior will be determined by both $p_{k_1,j_1}^{I'}$ and $p_{k_2,j_2}^{I'}$. Recall that $Eff_{g_{k,j}} \in [0, 1]$ is the efficacy of the guide $j$ targeting gene $k$. More specifically, by (1), (2) and (8), cell growth on a SKO or DKO depends on theoretical phenotypes $p_1$, $p_2$ that determine the growth when $Eff_{g_{k,j}} = 1$. Equation (13) above decreases the cell growth rate of cells infected with low-efficacy guides compared to the cells infected with high-efficacy guides. For $SKO(k,j)$ let $p_1 := p_{k,j}^{I'}$ and $p_2 := 0$, and for $DKO(k_1,j_1;k_2,j_2)$ let $p_1 := p_{k_1,j_1}^{I'}$ and $p_2 := p_{k_2,j_2}^{I'}$, in (7) to compute $\mathbf{p_y}' = (p_0^{y'}, p_1^{y'}, p_2^{y'}, p_3^{y'})$ to simulate the cell division behavior. In every cell doubling cycle, the cell will divide $y$ times with $y$ multinomial with $\mathbf{p_y} := \mathbf{p_y}'$ in (4).

**Computation of Simulation True Genetic Interaction**. With the initialized cell library specified above, and based on the methodology demonstrated in the cell-behaviors multinomial distributions sub-section, the simulated truth cell behavior with interactions of genes $k_1$ and $k_2$ was defined as follows:

1. Using $p_1 := p_{k_1}$ and $p_2 := p_{k_2}$ in (7), we computed the cell division probability based on KO gene effects **without** GI
   $\mathbf{p_y^0} := (p_0^{y_0}, p_1^{y_0}, p_2^{y_0}, p_3^{y_0})$

2. Using $p_1 =: p_{k_1}^I$ and $p_2 := p_{k_2}^I$ in (7), we computed the cell division probability based on KO gene effects **with** GI
   $\mathbf{p_y} := (p_0^y, p_1^y, p_2^y, p_3^y)$

3. The gene-level GI values were computed by plugging in the values $\mathbf{p_y^0}$ and $\mathbf{p_y}$ in (9).

The GIs were categorized as either negative, none, or positive. In the absence of an interaction $I_{k_1,k_2} := 0$, $\mathbf{p_y^0} = \mathbf{p_y}$ and (9) produces a 0 interaction. For example, the interaction of genes 1 and 2 in the toy example in S2 and S3 Tables $p_1 = 0.4$, $p_2 = -0.03$ and $\mathbf{p_y^0} = (p_0^{y_0}, p_1^{y_0}, p_2^{y_0}, p_3^{y_0}) = (0.030, 0.582, 0.388, 0)$ in item 1 above; $p_1 := p_1^I = 0.27$, $p_2 := p_2^I = 0.01$ and $\mathbf{p_y} = (p_0^y, p_1^y, p_2^y, p_3^y) = (0, 0.723, 0.275, 0.003)$ in item 2 above; producing a simulation true interaction of -0.1506 in item 3.

## Simulation system design II: Cell population transfection and selection

This subsection profiles the methods used for simulating cell population transfection and selection, and simulation data processing.

**Transfection and selection.** In the simulated cell population transfection and selection stage, we used $t$ as a cell doubling cycle counter and $i_e$ as a counter of bottleneck encounters. We initialized $t = 0$ and $i_e = 0$. The number of bottleneck encounters, $n_e$ was specified by the user, and the maximum number of cell doubling cycles was set to 30. The setup was as follows:

1. At $t = 0$, we initialized the current library of cell counts $c_1$ for every construct equal to the initial library; $c_1^{g_{k,j}} := c_0^{g_{k,j}}$ for SKO and $c_1^{g_{k_1,j_1;k_2,j_2}} = c_0^{g_{k_1,j_1;k_2,j_2}}$ for DKO. The $n_c$ dimensional vector of cell counts $\mathbf{C_1}$ was calculated by binding the current counts of SKO and DKO constructs as follows:

$$(c_1^{g_{k,j}} : k = 1, \ldots, n; j = 1, \ldots, n_g)$$

and

$$(c_1^{g_{k_1,j_1;k_2,j_2}} : k_1 = 1, \ldots, n-1; k_2 = k_1 + 1, \ldots, n; j_1, j_2 = 1, \ldots, n_g)$$

2. The parameters were stored to the $n_c \times 4$ matrix $M_p$, wherein row $i$ contains the cell division probability vector $\mathbf{p_y}' := \left( p_0^{y'}, p_1^{y'}, p_2^{y'}, p_3^{y'} \right)$ for construct $i$. Hence, construct $i$ has current counts $(\mathbf{C_1})_i$ and the cell division vector equal to row $i$ of $M_p$.

When $i_e < n_e$ and $t < 30$, we iteratively run:

1. Compute the current library size $D = \sum_{i=1}^{n_c} (\mathbf{C_1})_i$ and check the bottleneck condition:

$D > n_b$?

where $n_b$ is the user-prespecified bottleneck size.

- If **yes**, let $i_e = i_e + 1$, and draw $n_r$ cells from the current cell library:

$$\mathbf{C}' \sim \text{MVHyper} \left( \mathbf{n} = \mathbf{C_1}, \ k = n_r \right)$$

(14)

and set the current value of $\mathbf{C_1}$ equal to $\mathbf{C}'$. Here $\text{MVHyper}(\mathbf{n}, \ k)$ denotes the multivariate hypergeometric distribution that indicates the balls drawn from each color when $k$ balls are extracted without replacement from a urn containing $(\mathbf{n})_i$ balls of color $i$, here $\mathbf{n}$ is a vector of dimension the number of different colors in the urn. If, for example, the current library has 50 $DKO(1, 1; 2, 1)$ cells, the new library will have at most 50 of these cells. Following CRISPulator, we do not model multiple infections and assume the simulated CRISPR screening is on a low multiplicity of infection (moi) [11]. By default, we chose moi $\lambda = 0.3$ and model it as a Poisson process during transfection to select the cells that have single transfection occurrence by

$$P(x = 1; Poisson(\lambda))$$

where $P\left(x = 1; Poisson(0.3)\right) \approx 22\%$ of the $\mathbf{C_1}$ is set to be $n_r$, the number of cells that we kept after reaching the bottle neck.

2. For each construct $i$, grow $(\mathbf{C_1})_i$ to $(\mathbf{C'_1})_i$ following the row $i$ of $M_p$ that defines the corresponding $i^{th}$ cell division probability vector $\mathbf{p_y}' := (p_0^{y'}, p_1^{y'}, p_2^{y'}, p_3^{y'})$ for both $c_1^{g_{k,j}}$ if $SKO(k, j)$, and $c_1^{g_{k_1, j_1; k_2, j_2}}$ if $DKO(k_1, j_1; k_2, j_2)$. The current value of $(\mathbf{C_1})_i$ is set to $(\mathbf{C'_1})_i$ to simulate the cell populations' transfection and growth. In math, before growth we have $(\mathbf{C_1})_i$ construct $i$ cells, with each cell dividing according to (6), producing $(\mathbf{C'_1})_i := c_0 \times 0 + c_1 \times 2 + c_2 \times 2^2 + c_3 \times 2^3$ cells after one growth cycle with

$$(c_0, c_1, c_2, c_3) \sim Multinomial \left( (\mathbf{C_1})_i, values = (0, 1, 2, 3), prob = \mathbf{p_y}' \right)$$

So that $c_0 + c_1 + c_2 + c_3 = (\mathbf{C_1})_i$. And set $(\mathbf{C_1})_i = (\mathbf{C'_1})_i$.

Increase the iteration counter from $t$ to $t = t + 1$.

**Simulation data processing.** After this iterative process (when we reach either $i_e = n_e$ or $t = 30$), the cell library at this final timepoint $t_2$ was denoted as $L_{guide}^2$. We set the cell counts of $L_{guide}^2$ referred as $\mathbf{C_2}$ equal to $\mathbf{C_1}$, and computed the total library size $D = \sum_{i=1}^{n_c} (\mathbf{C_2})_i$. The relative frequency of each construct at $t_2$ was calculated as follows:

$$f_{g_{k,j}}^{rel'} = \frac{c_2^{g_{k,j}}}{D} \quad \text{for } SKO(k, j) \qquad f_{g_{k_1, j_1; k_2, j_2}}^{rel'} = \frac{c_2^{g_{k_1, j_1; k_2, j_2}}}{D} \quad \text{for } DKO\left(k_1, j_1; k_2, j_2\right)$$

(15)

where $k, k_1, k_2 \in \{1, \ldots, n\}$, $k_1 < k_2$ and $j, j_1, j_2 \in \{1, \ldots, n_g\}$.

Accordingly, we defined the $n_c$ dimensional vector of the relative frequency of constructs at $t_2$, $\mathbf{f}_{C_2}^{rel}$, by binding the relative frequency of SKO and DKO constructs at $t_2$ as shown below:

$$(f_{g_{k,j}}^{rel'} : k = 1, \ldots, n; j = 1, \ldots, n_g)$$

and

$$(f^{\text{rel}'}_{g_{k_1,j_1;k_2,j_2}} : k_1 = 1, \ldots, n-1; k_2 = k_1 + 1, \ldots, n; j_1, j_2 = 1, \ldots, n_g)$$

Similarly, using equation (12), we also defined the $n_c$ dimensional vector of the relative frequency of constructs at the initial timepoint $t_0$ $f^{\text{rel}}_{C_0}$ by binding the relative frequency of SKO and DKO constructs at $t_0$ as shown below:

$$(f^{\text{rel}}_{g_{k,j}} : k = 1, \ldots, n; j = 1, \ldots, n_g)$$

and

$$(f^{\text{rel}}_{g_{k_1,j_1;k_2,j_2}} : k_1 = 1, \ldots, n-1; k_2 = k_1 + 1, \ldots, n; j_1, j_2 = 1, \ldots, n_g)$$

Based on the definition of log fold change and $C_{pseudo}$ included in the parameters specification subsection, we calculated the Log2 $n_c$-dimensional Fold-Change (LFC) vector at $t_2$ vs. $t_0$ as follows:

$$\text{LFC} = \log_2 \left( \frac{f^{\text{rel}}_{C_2} + C_{pseudo}}{f^{\text{rel}}_{C_0} + C_{pseudo}} \right)$$

(16)

where $C_{pseudo}$ is the pseudo-count added to each relative frequency of the constructs to avoid $-\infty$ in logarithm. The rest of the sections in simulation system design are detailed in S1 Text, Supplemental Methods and Materials.

### Algorithmic designs: Monte-Carlo simulation on large scales

*Algorithm*: *Double CRISPR-Knockout Simulation (DKOsim)*. The practical computational workflow of DKOsim for large-scale applications is summarized in S1 Text, Simulation Steps, and the computational modules are presented in S1 Fig. Using the notations defined in the parameter specifications subsection, we initialized the simulated cell library as desired by users. For the purpose of modeling the competitive exponential cell growth and selection procedures after library transfection and transduction, we designed Algorithm 1 (Fig 4) to simulate the cell population adaptations through multinouli-resampling induced by the growth bottleneck.

### Results

#### DKOsim is tunable to infer the asymptotic effects of laboratory CRISPR screening parameters

We systematically ran DKOsim to investigate the tunability of the scheme and validate its use for simulating the empirically expected CRISPR screening data pattern from laboratory experiments. From the tunable parameters summarized in Table 1, we chose six parameters to compare the simulation tunability, including coverages, percentage of high-efficacy guides, GI magnitude, dispersion of the initial frequency of SKO counts, number of guides per gene, and cell doublings. To quantify the association between the GI identifications vs. simulated GIs, we applied Delta Log fold Change (dLFC) that calculates the deviations in log fold change (LFC) from the mean LFC of all constructs targeting gene pairs to the expectation of the sum of the single mutant fitness (SMF) for the two genes [19], and measured the association by Pearson's correlation r. Additionally, we calculated the precision and recall at 80 strongest dLFC GI identifications among the top 100 negative simulated GI hits (Precision/Recall@80) to quantify the dLFC GI identification performance. Since we were simulating both SKO and DKO, we applied BAGEL [20,21] to the simulated SKO data to assess the validity of the simulated essential genes, defined as genes with negative theoretical phenotypes in our designed system. Except for the tuning parameter being compared, the others were assumed to be the same for comparison. Specifically, for the simulated screenings that were compared, we summarized the default parameters of the baseline screening - Simulation

---

**Algorithm 1** DKOsim: Double CRISPR-Knockout Simulation Scheme via Monte-Carlo samplers*

---

1: **Inputs:** Number of genes $n$; Coverage $C$; Number of guides per gene $n_g$; % and magnitude of genetic interactions $\%_{GI}$ and $\sigma_{GI}$; % of each gene class; $\mu_{neg}$, $\mu_{pos}$, $\sigma_{neg}, \sigma_{pos}, \sigma_{WT}$ ; % of high-efficacy guides $\%_{heg}$; Number of bottleneck encounters $n_e$; Size of the bottlenecks $n_b$.

2: **Initialize** cell library with size $L_0 = n_c \times C$, where $n_c = \frac{n \times n_g \times (n-1) \times n_g}{2} + n \times n_g$, containing all possible combinations of $n$ genes and set interacting gene pairs by genetic interaction indicator $I_{k_1,k_2} = 1$ generated by randomly selecting $\%_{GI}$ of $DKO(k_1; k_2)$ without replacement.

3: **while** current number of cell doublings $i_e < n_e$ **do**

4: **if** Current cell library size $> n_b$ **then**

5: Cell growth reaches bottleneck, add one to bottleneck counter: $i_e = i_e + 1$. Subsample the cell library via multivariate-hypergeometric draw to select $Poisson(1 \mid \lambda = 0.3) \approx 22\%$ of $n_b$ cells, this is sampling cells without replacement from current cell library to $n_r = Poisson(1 \mid \lambda = 0.3) \times n_b$ cells.

6: **end if**

7: **for** each cell $o$ in current cell library, let $k = k(o) \in \{1, \ldots, n\}$ index SKO gene in the cell $o$, targeted by guides $j \in \{1, \ldots, n_g\}$; and let $k_1 = k_1(o) < k_2 = k_2(o) \in \{1, \ldots, n\}$ index DKO gene in the cell $o$, targeted by guides $j_1, j_2 \in \{1, \ldots, n_g\}$ **do**,

8: **if** cell is in $SKO(k(o), j)$ **then**

9: cell divides $x \in \{0, 1, 2\}$ times according to its phenotype $p_k \in [-1, 1]$ with $x$ following $Multinoulli((-p_k)_+, 1 - |p_k|, (p_k)_+)$, yielding either 0,2 or 4 descendant cells.

10: **else if** cell is in $DKO(k_1(o), j_1; k_2(o), j_2)$ **then**

11: cell divides $y \in \{0, 1, 2, 3\}$ times according to $\mathbf{p_y}' = (p_0^{y'}, p_1^{y'}, p_2^{y'}, p_3^{y'})$ with $y$ following $Multinoulli(p_0^{y'}, p_1^{y'}, p_2^{y'}, p_3^{y'})$, yielding 0, 2, 4 or 8 descendants.

12: **end if**

13: **end for**

14: One full iteration of the cell doubling cycle has been completed, add one counter to the cell doubling cycle counter: $t = t + 1$.

15: **end while**

16: **Return** the grown cell library after all pre-specified cell population growth, transfection, and selection

---

* Empirically, in large-scale simulation runs, we set a maximum of 30 doublings ($t \leq 30$) with moi $\lambda = 0.3$. The theoretical design of the Algorithm is summarized.

**Fig 4. Algorithm for double CRISPR knockout simulation.** Algorithmic design of Double CRISPR-Knockout Simulation (DKOsim) via Monte-Carlo samplers.

(Systematic Run - Baseline) in Table 2. One simulation was run for each unique combination set of parameters, and we visualized the analytical outputs in line diagrams (Fig 5).

For experimental parameters, including coverage, guide quality, initial constructs' counts dispersion, and cell doublings parameters, which can be directly controlled in the experimental design, we compared and visualized the effects of each by systematically running DKOsim (Fig 5A-5K). When increasing the coverage of the CRISPR screening experiments, the Pearson correlation r between the detected GI vs. the simulated GI monotonously increased and reached an asymptote around 0.7 at 100x (Fig 5A). Results of AUC-PR from BAGEL SKO gene essentiality identification further demonstrated that the simulated SKO essential genes asymptotically reach the optimal performance starting at 100x screening with 0.98 AUC-PR (Fig 5B). We quantified the GI identification performance of dLFC and visualized the changes in Precision/Recall@80 presented in Fig 5C, while the changes in the AUC-PR for all negative GIs are shown in S2A Fig. Monotone increasing patterns are found in both, with the asymptote at 100x. the Precision@80 dominates the Recall@80 and AUC-PR, reaching an asymptote around 0.86 at 100x. Beyond this 100x asymptotic point, increments to the experimental coverage do not significantly improve either the association or identification performance between dLFC vs. simulated GI, indicating the optimal cost-effective design for screening coverage is 100x.

**Table 2. Input parameters[1] in DKOsim simulation (Systematic Run - Baseline).**

| Parameters | |
|---|---|
| $n$ | 120 |
| $C$ | 100x |
| $n_g$ | 3 |
| %GI | 3% |
| $\sigma_{GI}$ | 1.5 |
| Percentage of each gene type: %neg | 15 |
| %pos | 5 |
| %wt | 75 |
| %ctrl | 5 |
| Theoretical phenotypes parameters: $\mu_{neg}, \sigma_{neg}$ | (-0.75, 0.1) |
| $\mu_{pos}, \sigma_{pos}$ | (0.75, 0.1) |
| $\sigma_{wt}$ | 0.25 |
| $\sigma_f$ | 1/3.29 |
| %heg | 100% |
| Guide-efficacy parameters: $\mu_{high}, \sigma_{high}$ | (0.9, 0.1) |
| $\mu_{low}, \sigma_{low}$ | (0.05, 0.07) |
| Mode: CRISPRn/CRISPRn-100%Eff | CRISPRn-100%Eff |
| Cell Doublings Parameters[2]: $n_b$ | 2 |
| $n_e$ | 1 |

1. We assume a maximum of 30 doubling cycles for each simulated experiment with moi $\lambda$=**0.3**.

2. The total number of cell doublings is 3.

We then compared the tunability for the percentage of high-efficacy guides. When the systematic runs were restricted to compare guide quality by tuning the high-efficacy guides percentages at 100x coverage, the correlations between dLFC and simulated GIs with increasing percentage of high-efficacy guides from 10% to 100% showed an increasing trend, with r=0.76 for 100% high-efficacy guides (Fig 5D). We additionally tested whether there are synergistic effects between high-efficacy guides percentage and coverage. Simulations were run on increasing percentage of high-efficacy guides from 10% to 100%, stratified by coverage at 10x, 100x, and 200x. Results showed that correlations increase with higher coverage overall, comparing the same level of percentage of high-efficacy guides. We observed the effect of increasing percentage of high-efficacy guides on correlation is synergistically enhanced by higher coverage, particularly at low-to-moderate efficacy ranges, where improvements in guide-efficacy translate into disproportionately larger gains in correlation.

Fig 5E shows BAGEL essentiality identification results: while BAGEL AUC-PR in recovering SKO gene essentiality is high (AUCPR>0.75) across almost all conditions, benefits of increasing high-efficacy guides percentage depend on coverage, with substantial improvements at low coverage and diminishing returns once coverage is sufficient. This suggests an interaction but weak-additive synergy interaction between high-efficacy guides percentage and coverage for SKO gene essentiality identification. To evaluate GI identification performance from dLFC, we examined changes in Precision/Recall@80 for the top-ranked negative genetic interactions across increasing percentage of high-efficacy guides (Fig 5F), as well as AUC–PR for all negative GIs (S2B Fig). Both precision and recall increased with guide efficacy; however, the

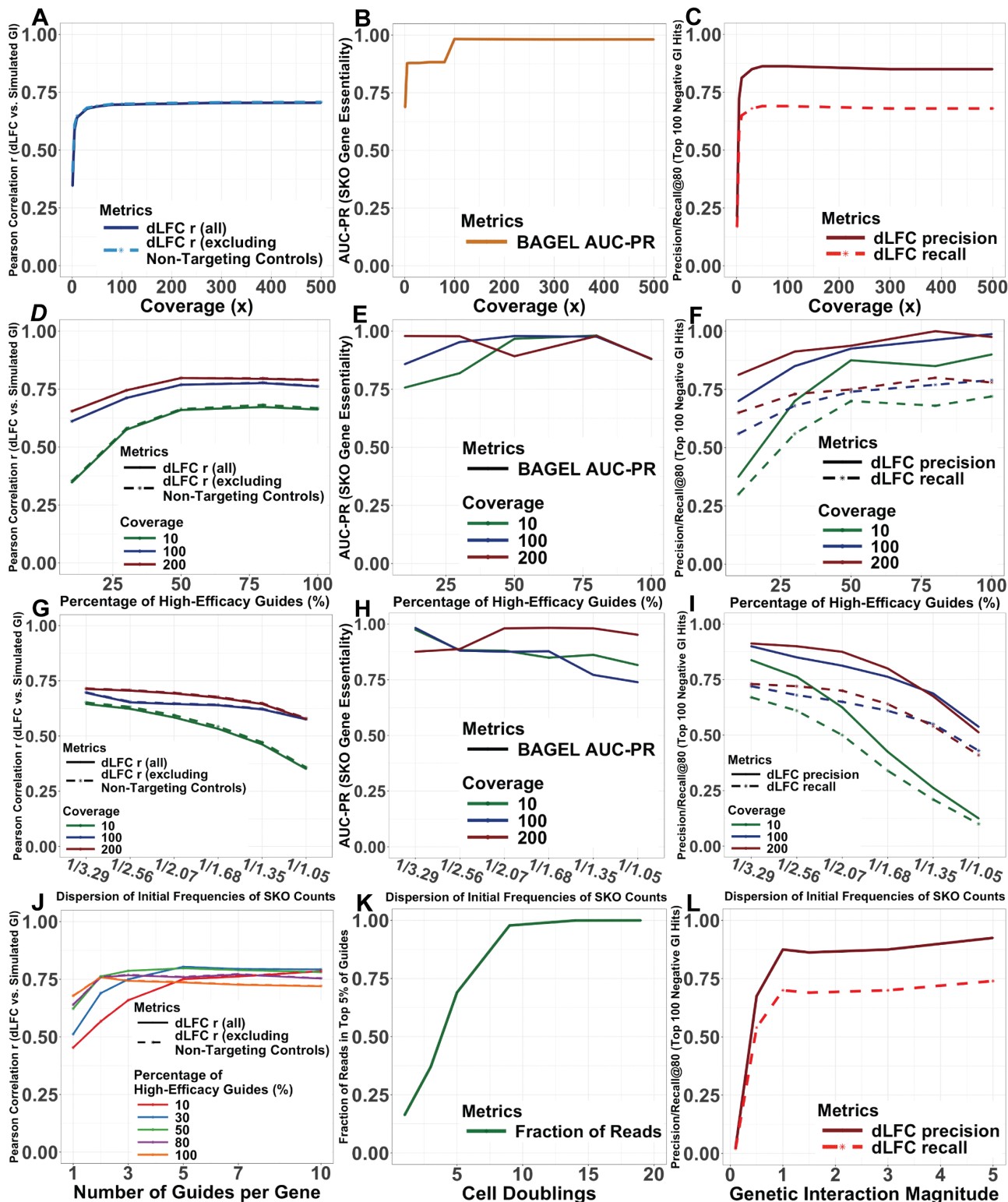

**Fig 5. Tunability of the parameters in the simulation scheme.** Delta Log Fold-Change (dLFC) is applied to simulations to measure the genetic inter-action (GI) scores. **(A)-(C)** Simulation Runs on Coverage Effects. We use the parameters in Table 2 except for the coverage C varying from 1x to 500x: **(A)** Changes of Pearson correlation r on dLFC vs. simulated GI, from Coverage varying runs. **(B)** Changes of AUC-PR for SKO Gene Essentiality on

BAGEL identifications on the simulated essential (negative phenotypic) genes, from Coverage varying runs. **(C)** Changes of Precision and Recall of the 80 most negative dLFC identifications on the top 100 simulated negative GI hits, from Coverage varying runs. **(D)-(F)** Simulation Runs on Percentage of High-Efficacy Guides Effects. We use the parameters in Table 2 except for the percentage of guides with high-efficacy %*heg* varying from 10% to 100% with Mode on "CRISPRn", stratified by coverage $C$ at 10x, 100x, and 200x: **(D)** Changes of Pearson correlation r on dLFC vs. simulated GI, from High-Efficacy Guides Percentage varying runs. **(E)** Changes of AUC-PR for SKO Gene Essentiality on BAGEL identifications on the simulated essential (negative phenotypic) genes, from High-Efficacy Guides Percentage varying runs. **(F)** Changes of Precision and Recall of the 80 most negative dLFC predictions on the top 100 simulated negative GI hits, from High-Efficacy Guides Percentage varying runs. **(G)-(I)** Simulation Runs on Dispersion of Initial Counts Effects. We use the parameters in Table 2 except for the dispersion of the initial frequency of SKO counts $\sigma_f$ varying from $\frac{1}{1.05}$ to $\frac{1}{3.29}$, stratified by coverage $C$ at 10x, 100x, and 200x: **(G)** Changes of Pearson correlation r on dLFC vs. simulated GI, from Initial Counts Dispersion varying runs. **(H)** Changes of AUC-PR for SKO Gene Essentiality on BAGEL identifications on the simulated essential (negative phenotypic) genes, from Initial Counts Dispersion varying runs. **(I)** Changes of Precision and Recall of the 80 most negative dLFC predictions on the top 100 simulated negative GI hits, from Initial Counts Dispersion varying runs. **(J)** Simulation Runs on Number of Guides per Gene Effects. We use the parameters in Table 2 except for the number of guides per gene $n_g$ tuned varying from 1 to 10 and the percentage of guides with high-efficacy %*heg* tuned from 10% to 100%: Changes of Pearson correlation r on dLFC vs. simulated GI, from Number of Guides per gene varying runs, colored by the percentage of high-efficacy guides. **(K)** Simulation Runs on Cell Doublings Effects. We use the parameters in Table 2 except for the bottleneck size $n_b$ and the number of bottleneck encounters $n_e$ tuned to control the cell doublings varying from 1 to 19: Changes in the fraction of reads in the top 5% of the guides, from Cell Doublings varying runs. **(L)** Simulation Runs on GI Magnitude Effects. We use the parameters in Table 2 except for the strength of the simulated GIs $\sigma_{GI}$ varying from 0.1 to 5: Changes of Precision and Recall of the 80 most negative dLFC identifications on the top 100 simulated negative GI hits, from GI Magnitude varying runs.

magnitude of improvement strongly depended on library coverage. Specifically, increases in the proportion of high-efficacy guides yielded the largest gains in both Precision@80 and Recall@80 under low-coverage conditions at 10x, whereas performance improvements diminished as coverage increased to 100x and 200x. At moderate to high coverage, Precision@80 approached 0.95, indicating that approximately 76 of the top 80 ranked interactions correspond to true simulated negative GIs. In contrast, recall improved more gradually and exhibited earlier saturation, reflecting the greater difficulty of exhaustively recovering all true interactions. Together, these results demonstrate a coverage-dependent, synergistic interaction between library coverage and guide efficacy, whereby improvements in guide efficacy disproportionately enhance GI detection performance when guide sampling is limited, with diminishing returns once coverage becomes sufficient.

To demonstrate the tunability of the initial counts dispersion, following Table 1, we systematically ran DKOsim by tuning $\sigma_f$ based on the z-score resulting from a 40–90% confidence level. For example, at 90% confidence, the expected z-score is 3.29, and we constructed the SKO gene frequency following a normal distribution with $\sigma_f = \frac{1}{3.29}$ so that there is a 10-fold difference between the 95th and 5th percentiles. As defined, with 100x coverage, lower confidence leads to higher $\sigma_f$, resulting in a higher dispersion of the initial count distribution. Based on this setting, we visualized the changes of the correlation r with increasing setting to the dispersion of initial frequencies of SKO counts (Fig 5G). The increments of the initial counts dispersion decrease the correlation r on dLFC vs. simulated GI overall, and we found that r drops to 0.58 when $\sigma_f = \frac{1}{1.05}$. On metrics of correlation r, we tested whether there are synergistic effects between the initial counts dispersion and coverage. Increasing dispersion in initial SKO counts' frequencies consistently reduced the correlation between dLFC and simulated GIs; however, the magnitude of this effect depended on library coverage. In particular, high dispersion caused a disproportionate loss of correlation under low-coverage conditions, whereas higher coverage partially buffered against uneven starting frequencies. These results indicate a synergistic interaction between library coverage and the initial SKO counts dispersion, whereby limited sampling and skewed guide abundances jointly amplify stochastic noise and degrade GI detection performance from dLFC.

We applied BAGEL essentiality identification on simulations in this setting. Increasing dispersion in initial SKO counts' frequencies had only a modest effect on SKO gene essentiality identification performance, with AUC-PR remaining high (AUCPR > 0.75) across most conditions (Fig 5H). While higher dispersion led to gradual performance degradation at low to moderate coverage, high coverage largely buffered against uneven starting frequencies, maintaining near-ceiling AUC-PR values. Unlike GI detection, SKO essentiality exhibited a saturating response to both coverage and initial frequency

dispersion, indicating limited interaction and no strong synergistic effect between these parameters. Increasing dispersion in initial SKO counts' frequencies led to a pronounced decline in both Precision@80 and Recall@80 for top negative GI detection (Fig 5I). Importantly, the magnitude of this degradation depended strongly on library coverage: under low-coverage conditions, even moderate dispersion caused a rapid collapse in both precision and recall, whereas higher coverage substantially mitigated these effects. The highly non-parallel performance trajectories indicate a strong synergistic interaction between coverage and initial frequency dispersion, whereby limited sampling and skewed guide abundances jointly amplify stochastic noise and severely compromise GI detection.

We further tested the effects of the number of guides per gene (Fig 5J) on different proportions (10%-100%) of high-efficacy guides. With only 10% highly efficient guides, we observed a monotone increasing trend of the Pearson correlation r on dLFC vs. simulated GI, peaking at 0.78 with 10 guides per gene. When 30% of the guides were highly efficient, 5 guides per gene with r = 0.8 is the optimum of the correlation; when either 50% and 80% of the guides were highly efficient, we observed an asymptotic optimum of the correlation at 3 guides per gene with 0.79 and 0.77, respectively; and when all of the guides were highly efficient, the correlation reached the optimum with r = 0.76 at 2 guides per gene and, unexpectedly, show a decreasing trend beyond this point, indicating dLFC might not identify the GI well in a perfect guide-efficacy scenario. Based on the simulation results, choosing 3 guides per gene would be enough in real laboratory screening by ensuring sufficient quality of the guides when ordering. Our simulation reflected the synergistic effects between high-efficacy guide rates and the number of guides per gene in CRISPR screens. Considering a correlation of at least 0.75 for GI to be effectively identified, for sets of the simulation with high-efficacy guides proportions lower than 50%, more guides for each target gene (≥ 5) need to be incorporated into library design. For experiments with higher proportions of high-efficacy guides (≥ 80%), two guides are generally sufficient. This shows that the number of guides per gene and high-efficacy guide proportion interact synergistically, with more guides set for each target yielding disproportionately greater stability when higher fractions of guides achieve strong knockout efficiency.

For cell doublings, we mainly compared and visualized the asymptotic trends of the fraction of reads in the top 5% reads for cell doublings in Fig 5K. From 1 to 19 cell doublings, the fraction of reads in the top 5% reads increased monotonously due to the decrease in cell diversity caused by the death of simulated cells with negative phenotypic targeted KO genes' constructs. With 19 doublings, 100% of the reads are represented by the top 5% of guides, indicating the cell diversity reaches a minimum where the cell counts are dominated by one specific construct type, possibly in cells with dual-positive combinatorial KO gene constructs with positive GIs.

For biological parameter, specifically GI magnitude, which is determined intrinsically by genetic characteristics in CRISPR screening and not directly controlled by the laboratory experimenter, we visualized its effects on Precision/Recall@80 of dLFC on the top 100 negative GI hits (Fig 5L). Results showed that precision and recall for the top hits reached the asymptotes when $\sigma_{GI}$ is 1, where Precision@80 is 0.875, Recall@80 is 0.7, supported by asymptotic AUC-PR as 0.792 (S2C Fig).

We prepared a summary of guidance on parameter selection for running simulations. The guideline can be accessed in "Summary Guidance in Picking Suitable Parameters" section from the tutorial in vignettes of DKOsimR, our built R package for DKOsim. Check Data Availability Statement to install and access DKOsimR.

## DKOsim approximates patterns from actual laboratory Double-CRISPR Knockout screening data

We compared the actual experimental data to the synthetic data approximated by our simulation. We collect three sets of laboratory screening data for approximation, including the combinatorial CRISPR-Cas9 screens designed by Shen et al.[4] (Shen-2017), "Big Papi" orthologous combinatorial CRISPR-Cas9 screens designed by Doench et al.[22] (Doench-2017), and "SCHEMATIC" combinatorial CRISPR platform to map synthetic lethal interactions designed by Fong et al.[23] (Fong-2024). In this section, the data approximation is restricted to the A549 cell line, a human lung adenocarcinoma cell line with a KRAS gain-of-function mutation in the oncogenic background of cancer studies.

We investigated whether DKOsim can approximate constructs' count distribution at the initial timepoint, given the same number of perturbed genes. To approximate Shen-2017 design[1], we initialized Simulation (mimicking Shen) by 120 uniquely perturbed single genes, 3 guides per targeted gene with 3% GIs, and set 80% confidence (expected z-score as 2.56) on dispersion of SKO genes $\sigma_f = \frac{1}{2.56}$ for DKOsim (Fig 6A). Comparing simulation results with Shen-2017 day 3 collected constructs' counts in the A549 cell line, histograms of the log 10 counts are highly aligned between the laboratory Shen-2017 vs. Simulation (mimicking Shen). To approximate the Doench-2017 design, we initialized Simulation (mimicking Doench) by 28 uniquely perturbed single genes, 5 guides per targeted gene with 3% GIs, and set 90% confidence (expected z-score as 3.29) on dispersion of SKO genes $\sigma_f = \frac{1}{3.29}$ for DKOsim (Fig 6B). Compared with Doench-2017 plasmid constructs' counts in the A549 cell line, histograms of the log 10 counts highly overlapped between the laboratory Doench-2017 vs. Simulation (mimicking Doench). To approximate the Fong-2024 design, we initialized Simulation (mimicking Fong) by 246 uniquely perturbed single genes, 3 guides per targeted gene with 3% GIs, and set 80% confidence (expected z-score as 2.56) on dispersion of SKO genes $\sigma_f = \frac{1}{2.56}$ for DKOsim (Fig 6C). Similarly, we compared simulation results with Fong-2024 plasmid constructs' counts in A549 cell line, histograms of the log 10 counts highly overlapped between the laboratory Fong-2024 vs. Simulation (mimicking Fong).

Based on the aligned distributions of constructs' counts at the initial timepoint, we investigated whether DKOsim could simulate actual cell growth and approximate real experimental data patterns on LFC. Specifically, we chose the Fong-2024 [23] design (data characteristics summarized in S4 Table) for simulation approximation, in which this recently developed combinatorial CRISPR platform comprises a panel of 246 genes with 67 frequently mutated genes, asymmetrically crossing another 176 druggable genes, with 3 additional non-targeting controls that do not affect the functions of the cells. Among these genes, 64 genes are treated as essential, where AAVS1 is known as a safe harbor locus that should not disrupt any cell function and is treated as the negative control. The cell line was infected at a multiplicity of infection (MOI) of 0.3 to ensure > 100x coverage in library production. Each gene was targeted by 3 independent guides in this asymmetric library. Within all possibly interacting 12282 gene pairs, 400 pairs were under FDR < 10%, and we treated 3% ($\sim \frac{400}{12282}$) of the gene pairs as truly interacting pairs. While screening, two biological replicates were included, each with independent viral transduction on low numbers of cell passages.

To approximate the laboratory design, in our Simulation (mimicking Fong) initialization (Fig 6D Parameters), we included 246 genes: 64 negative phenotypic genes to align with the number of essentials, 178 wild-type genes to align with the number of nonessentials, and 4 non-targeting controls to align with the AAVS and 3 non-targeting controls from Fong-2024. Each gene was targeted by 3 independent guides, and the asymmetric design in the lab was extended to a symmetric library where all the initialized genes could interact with each other. To further mimic the design, we set coverage $C = 1000x$ with MOI = 0.3 to align with the high laboratory coverage, and 3 times cell doublings to align with the low number of passages. We simulated 2 biological replicates, each transduced independently. LFC was calculated for each replicate and aggregated by mean in the final output after all simulated cell growth, transfection, and selections to approximate Fong-2024's data pattern.

We compared the overall LFC distributions from the Fong-2024 versus the A549 cell line simulation (Fig 6D). Within the same range, DKOsim Simulation (mimicking Fong) approximates Fong-2024's LFC pattern with almost perfect overlaps. To compare GI distributions, we applied dLFC to Simulation (mimicking Fong) to identify the simulated GIs, followed by z-standardization to the identification scores named zdLFC. Comparing distributions of zdLFC with SCHEMATIC interaction scores from Fong-2024 (Fig 6E). The overall shape is aligned and most interaction scores fall within -2.5 to 2.5, zdLFC approximates SCHEMATIC scores with merely rightward distribution and a few more spikes, possibly due to the inclusion of the simulated positive GI in DKOsim. In contrast, SCHEMATIC is specifically designed for identifying actionable synthetic lethal (negative) interactions.

We then compared LFC distributions for its unique gene combinations. Utilizing the 64 gene essentiality labels from Fong-2024, we deconvoluted both laboratory LFC and simulated LFC by unique gene combos. To align with laboratory design, we treated our simulated negative genes as essentials, kept the non-target controls with same number as

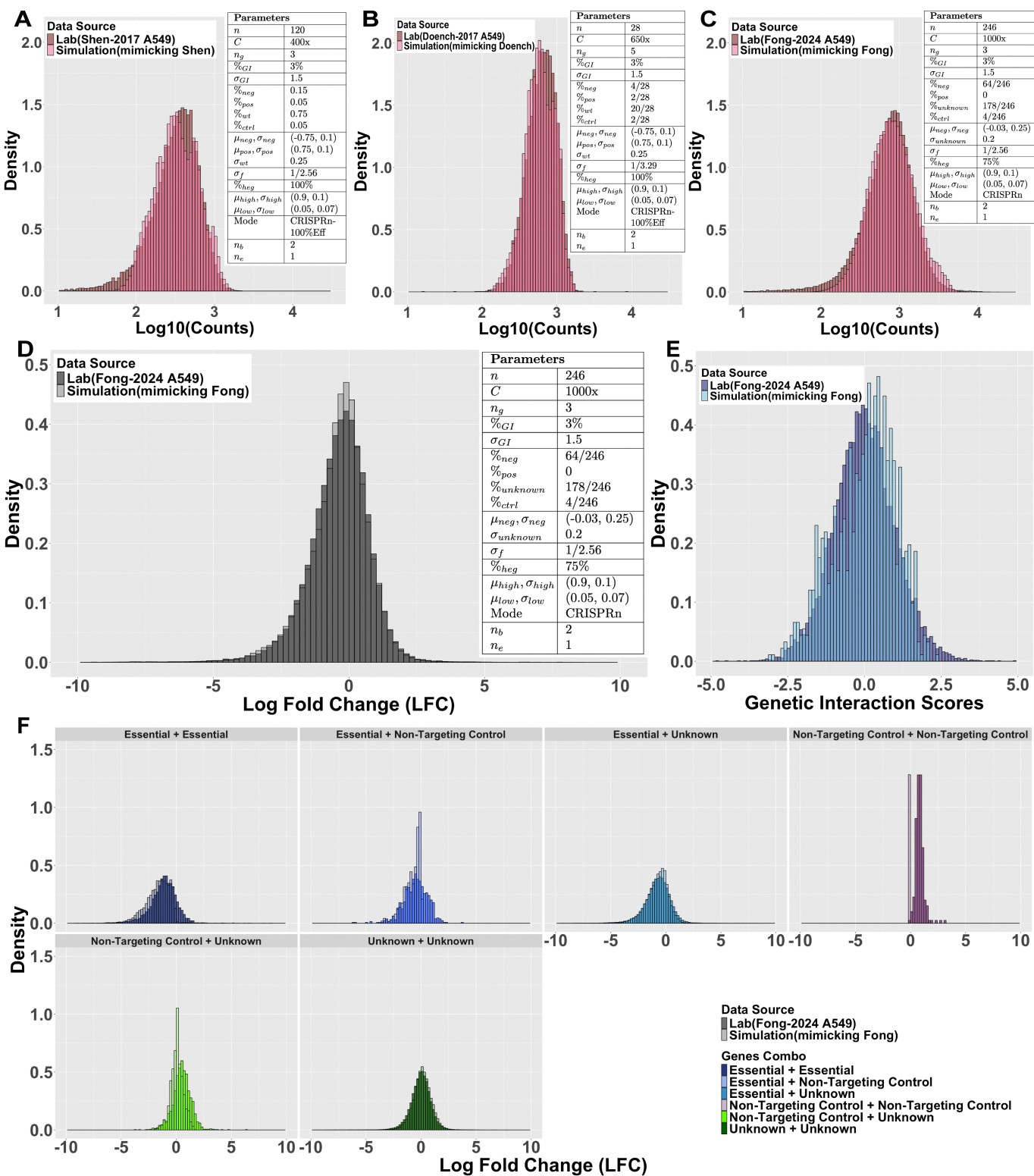

**Fig 6. Simulation approximation on laboratory data.** Comparison of Distributions between Simulation and Laboratory Data. A maximum of 30 doubling cycles for each simulation with moi $\lambda = 0.3$ is assumed. **(A)** Histograms of log10-scaled constructs' counts at the initial timepoint on Shen-2017 A549 vs. Simulation Run (mimicking Shen). Parameters table indicates values of each input parameter for DKOsim (mimicking Shen-2017 A549)

and the total number of simulated cell doublings is 3. **(B)** Histograms of log10-scaled constructs' counts at the initial timepoint on Doench-2017 A549 vs. Simulation Run (mimicking Doench). Parameters table indicates values of each input parameter for DKOsim (mimicking Doench-2017 A549) and the total number of simulated cell doublings is 3. **(C)** Histograms of log10-scaled constructs' counts at the initial timepoint on Fong-2024 A549 vs. Simulation Run (mimicking Fong). Parameters table indicates values of each input parameter for DKOsim (mimicking Fong-2024 A549) and the total number of simulated cell doublings is 3. To associate gene labels from lab data, we initiate the theoretical phenotypes $p_k$ in unknown gene class by $p_k \sim N_{[-0.5, 0.5]}(\mu_{unknown} = 0, \sigma^2_{unknown})$. **(D)** Histograms of overall LFC distributions on Fong-2024 A549 vs. Simulation Run (mimicking Fong). Parameters table indicates values of each input parameter for DKOsim (mimicking Fong-2024 A549) and the total number of simulated cell doublings is 3. To associate gene labels from lab data, we initiate the theoretical phenotypes $p_k$ in unknown gene class by $p_k \sim N_{[-0.5, 0.5]}(\mu_{unknown} = 0, \sigma^2_{unknown})$. **(E)** Histograms of genetic interaction (GI) scores from Schematic on Fong-2024 A549 vs. zdLFC on Simulation (mimicking Fong). **(F)** Histograms of LFC by gene-gene combinations on Fong-2024 A549 vs. Simulation (mimicking Fong).

Fong-2024, and categorized the rest of the simulated genes, including wildtype and positive genes, as unknown. The deconvolution results (Fig 6F) demonstrated that LFC of the simulated co-essential genes, co-unknown genes, and essential with unknown combos are well approximating the laboratory data. The trend of alignment is most prominent among the co-unknown genes representing the predominant construct category in Simulation (mimicking Fong), where we found almost perfect alignment between the simulation and laboratory data. However, LFC for gene combos consists of the simulated non-targeting control genes tending to have smaller variability compared to Fong-2024, mainly due to our strict definition of non-targeting controls in having 0 theoretical phenotypes. Since the non-targeting controls are explicitly defined to not affect the exponential cell growth in DKOsim, this results in markedly reduced variability in simulation, compared to actual laboratory experimental data.

### DKOsim simulates the noise of existence from laboratory experimental replicates and is reproducible from randomness

While DKOsim is applicable to approximate data from laboratory CRISPR experiments in real-world settings, we ensured the design of this simulation system to achieve a high degree of reproducibility. Reproducibility, as defined in our context, primarily encompasses two aspects: DKOsim is reproducible to maintain both stochastic consistency across Monte-Carlo randomizations and a sense that users across many disciplines can obtain consistent simulation results.

We measured the reproducibility of DKOsim using Pearson correlation r between two replicates of simulations in DKOsim, on the asymptotic effects of the experimental parameters coverage, high-efficacy guides percentage, and initial counts dispersions. For coverage from 1x to 500x, DKOsim asymptotically gained higher reproducibility between replicates, shown by the increasing correlations, where at 100x, the correlation asymptotically approached 0.95 and reached 0.99 up to 500x (Fig 7A). A monotone increasing trend was seen in the replicates' reproducibility as the percentage of high-efficacy guides increased from 10% to 100%. When all guides are 100% efficient in knocking out the target genes, the correlation between the replicates was 0.87, indicating a highly consistent LFC between the two replicates (Fig 7B), as empirically expected, improved guide quality is contributing to greater reproducibility of the experiments. We observed a monotone decreasing trend in the replicates' reproducibility, given a larger dispersion of initial counts (Fig 7C), with correlation r dropping from 0.95 to 0.91 with higher dispersion.

We compared the reproducibility between the laboratory experimental replicates and the simulated replicates in DKOsim across different experimental screening designs. We additionally collected laboratory data from the combinatorial CRISPR-Cas9 metabolic screens designed by Zhao et al.[24] (Zhao-2018), and the "in4mer" CRISPR-Cas12a multiplex knockout screens designed by Hart et al.[6] (Hart-2024). Following the original combinatorial CRISPR screening design of Shen et al.[4], Zhao-2018 and Fong-2024 [23] reproduced the dual-vectorized DKO setup with independent lentiviral transductions per context, where each replicate was independently initiated to ensure reproducibility. This approach offers greater flexibility in scaling the interacting gene pool and captures full biological variability signals in replication from guide integration and infection noise, but introduces more stochastic dropouts depending on guide quality, and higher labor

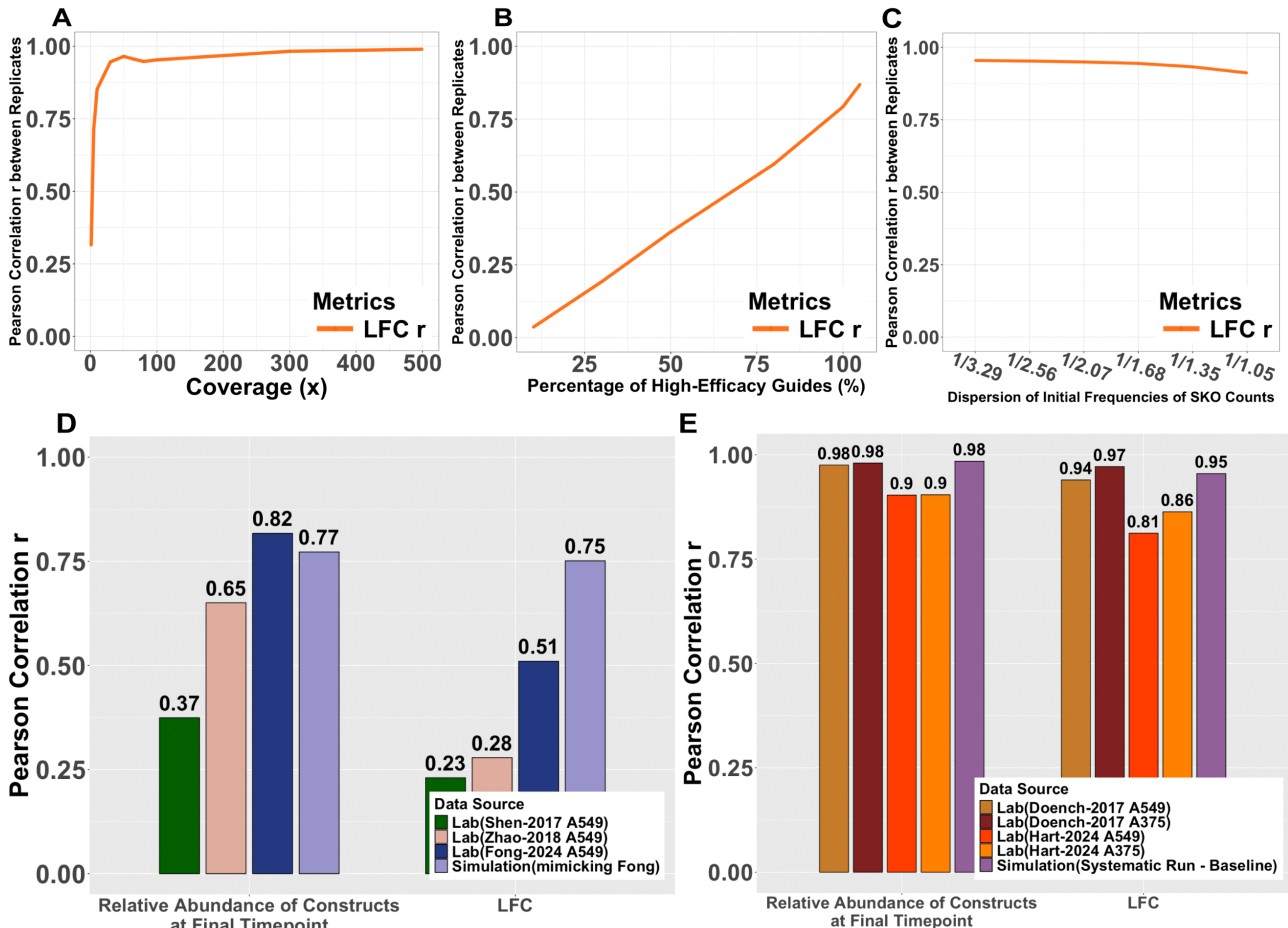

**Fig 7. Reproducibility of the simulated CRISPR experiments. (A)-(C)** Systematic Tunability Effects on the simulation reproducibility: **(A)** Changes of Pearson correlation r on LFC between two replicates of the simulation runs, from Coverage varying runs. **(B)** Changes of Pearson correlation r on LFC between two replicates of the simulation runs, from High-Efficacy Guides Percentage varying runs. **(C)** Changes of Pearson correlation r on LFC between two replicates of the simulation runs, from Initial Counts Dispersion varying runs. **(D)-(E)** Comparison of reproducibility on dual-CRISPR screening experiments between laboratory and simulation data: **(D)** Barplots of Pearson correlation r on the relative frequency of guides and LFC between laboratory replicates with dual-vector designs on two independent viral transduction vs. Simulation (mimicking Fong). **(E)** Barplots of Pearson correlation r on the relative frequency of guides and LFC between laboratory replicates with single-vector design on one pooled-viral transduction vs. Simulation (Systematic Run - Baseline).

complexity from dual transduction in practical experimental runs. We compared the correlation of the replicates with the dual vectorized design using Simulation (mimicking Fong) which was designed to approximate the Fong-2024 A549 data (Fig 7D). Two sets of the correlations were compared, first on the relative abundance of constructs at final timepoint, and on the LFC between final timepoint vs. initial timepoint from independent transductions. We found increasing replicate correlations from relative abundance compared to the original design of Shen-2017, and the close agreement with Fong-2024 implies that the simulation accurately models and captures the biological noise arising from cell growth and the selection process. While DKOsim approximated the noise signals from replicates, it maintained the highest LFC correlations when compared with other dual-vectorized DKO laboratory data.

Both Doench et al. and Hart et al. built single-vector lentiviral delivery design. Doench-2017 [22] design encoded the dual-sgRNA in a single lentiviral construct and conducted one viral transduction per replicate, yielding high precision and

reproducibility with cleaner delivery but sacrificing true biological independence. In contrast, Hart-2024 design had one lentiviral construct encoding up to 4 guide RNAs using Cas12 with one-time viral transduction in the pooled in4mer library. Within the same transduction, multiple sequencing was conducted to measure the constructs' counts, resulting in two technical replicates at the final timepoints. This approach ensures high precision and reproducibility on pre-defined KO gene combos and minimizes the possible dropout and noise. But as with the Doench design, the one-time lentiviral transduction predisposes the replicates' reproducibility towards the technical consistency rather than full biological independence. Under this experimental design of a single vector, we compared the replicates' correlation with Simulation (Systematic Run - Baseline) designed to systematically infer the optimal CRISPR library designs (Fig 7E). Two sets of correlations were compared: first on the relative abundance of constructs at the final timepoint, and on the LFC between final timepoint vs. the plasmid library. Results showed that while all screening yielded strong reproducibility, replicates' correlation closely agrees with Hart-2024 and is identical to Doench-2017, indicating that DKOsim can also be utilized to capture noises in single-vectorized DKO design for both biological and technical replicates. This conclusion is further supported by the LFC correlations, where simulated LFC correlates better than the technical replicates from Hart-2024 and falls between the Doench-2017 A549 and A375 cell lines.

## Discussion

GIs are rare, and many challenges remain in systematically profiling them without the possibility of validating all gene pairs in high-throughput experiments and in constructing gold standard datasets with true interaction values. On the one hand, laboratory experimental scientists invest a lot of time and resources in performing multiplexed CRISPR screening, which presents both biological and technical difficulties. On the other hand, many existing computational tools have devoted much effort to minimizing the CRISPR screening data noise in order to perform the most robust estimation of gene-gene interactions. But without the underlying truths of the GI values, and systematic quantitative views of the experimental CRISPR screening schemes and parameters, it is only possible to validate the partial interaction detections in restricted cell lines by gathering a large amount of screening data across multiple platforms with different library designs and varying data quality, resulting in a tremendous amount of uncertainty.

To address the aforementioned problems, we designed a Double-CRISPR Knockout Simulation Scheme (DKOsim), which systematizes a Monte-Carlo simulation methodology applicable to mimic both the SKO and DKO laboratory CRISPR knockout screening experiments and data patterns, while ensuring high tunability and reproducibility. Utilizing DKOsim on a large scale, users can simulate desired CRISPR screening datasets with the underlying true values of GIs as input, which serves the goals of both inferring the optimal experimental design for CRISPR knockout screening and supporting the statistical rigor in method development to perform inference on interaction values.

Accordingly, to better demonstrate the working logics of our designed scheme, we probabilistically derived the cell growth behaviors' distributions for both SKO and DKO cells, and defined simulated GIs based on growth rates of the cells. We incorporated the guide-efficacy design into our simulation and summarized all tunable components in DKOsim. To make the large-scale implementation practically feasible, we designed an algorithm for DKOsim to compile the entire theoretical framework in generating a simulated cell population that mimics the practical laboratory experimental process. As evident from the simulation results, the approximation to data patterns from practical laboratory experiments shows superior alignments between the simulation and laboratory data in many ways, and as expected, the distribution of the simulated GI values recapitulates the laboratory-derived interaction scores. This shows the unprecedented potential of applying DKOsim to generate analyzable synthetic data for downstream analysis.

Furthermore, DKOsim framework was designed to achieve both stochastic consistency and real-world fidelity. Specifically, it aims to reproduce stable and coherent outcomes across parallelly repeated runs, despite the inherent randomness of Monte Carlo processes, while also capturing the biological and technical variability characteristic across many CRISPR screening experimental designs. In addition to modeling experimental noise realistically, DKOsim emphasizes user-level reproducibility: the platform provides a modular simulation R package named DKOsimR, ensuring future researchers from

diverse backgrounds can reliably reproduce results by following the same defined workflow and tutorials. This dual repro-ducibility, emphasized on algorithmic robustness and user consistency, makes DKOsim a versatile tool for benchmarking and evaluating CRISPR-based double-knockout screening strategies.

Limitations exist in current schemes, which will be addressed in future work. First, we are only modeling the on-targeting effects of the screening. The off-target effects might be an important consideration to incorporate into current schemes. Second, we simplified the cell growth assumptions for derivations and simulation programming; the cells may divide more than twice in unequal time intervals. Taking continuous time effects into current model might reflect more signals in approximating true biological variability. Third, we mainly rely on Pearson correlation coefficient to validate DKOsim's reproducibility but this might not consider context difference when comparing to real experimental data. Future comparisons on context-specific replicates need to take involved for a through comparison between simulations and real datasets. Lastly, DKOsim is mainly designed to simulate DKO CRISPR screening, though it also simulates SKO effects. Its scalability to increase the number of initialized single genes is not perfected at the current stage.

## Supporting information

**S1 Text. Supplemental methods and materials.**
(DOCX)

**S1 Fig. Overview of double-CRISPR knockout simulation schematic workflow.** Created in BioRender. Gu, Y. (2026) https://BioRender.com/93cmnmy.
(TIF)

**S2 Fig. AUC-PR for all Negative GIs on asymptotic runs of dLFC identifications.** Delta log fold-change (dLFC) is applied to simulations to measure the genetic interaction (GI) scores. Simulated GIs are restricted to negative values for calculating AUC-PR. Line plots show the changes of AUC-PR for all simulated negative GIs on asymptotic runs of: **(A)** Coverage; **(B)** Percentage of high-efficacy guides effects; **(C)** GI Magnitude; **(D)** Confidence level for initial counts disper-sion (%); **(E)** Number of guides per gene.
(TIF)

**S1 Table. Toy example: Single Gene KO parameters.**
(DOCX)

**S2 Table. Toy example: Single & Double KO Genes Initialization.**
(DOCX)

**S3 Table. Toy Example: Initial cell library.**
(DOCX)

**S4 Table. Summary Table for Fong 2024 data characteristics.**
(DOCX)

**S1 Data. Statistics of DKOsim systematic runs.**
(XLSX)

**S2 Data. DKOsim Run 95 (Systematic Run - Baseline) data with dLFC scores.**
(ZIP)

**S3 Data. DKOsim Run 139 (mimicking Fong 2024) data with dLFC scores.**
(CSV)

## Author contributions

**Conceptualization:** Yue Gu, John Paul Shen.

**Data curation:** Yue Gu, Luis Novelo.

**Formal analysis:** Yue Gu.

**Funding acquisition:** John Paul Shen.

**Investigation:** Yue Gu, Traver Hart, Luis Novelo, John Paul Shen.

**Methodology:** Yue Gu, Luis Novelo.

**Resources:** Yue Gu, Traver Hart, John Paul Shen.

**Software:** Yue Gu.

**Supervision:** Traver Hart, Luis Novelo, John Paul Shen.

**Validation:** Yue Gu, Traver Hart, Luis Novelo, John Paul Shen.

**Visualization:** Yue Gu, Traver Hart, John Paul Shen.

**Writing – original draft:** Yue Gu.

**Writing – review & editing:** Yue Gu, Traver Hart, Luis Novelo, John Paul Shen.

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
