## [Decision Letter · Decision Letter 0]

10 Nov 2025

PCOMPBIOL-D-25-01827

Double-CRISPR Knockout Simulation (DKOsim): A Monte-Carlo Randomization System to Model Cell Growth Behavior and Infer the Optimal Library Design for Growth-Based Double Knockout Screens

PLOS Computational Biology

Dear Dr. Shen,

Thank you for submitting your manuscript to PLOS Computational Biology. After careful consideration, we feel that it has merit but does not fully meet PLOS Computational Biology's publication criteria as it currently stands. Therefore, we invite you to submit a revised version of the manuscript that addresses the points raised during the review process.

Please submit your revised manuscript within 60 days Jan 10 2026 11:59PM. If you will need more time than this to complete your revisions, please reply to this message or contact the journal office at ploscompbiol@plos.org. Please include the following items when submitting your revised manuscript:

We look forward to receiving your revised manuscript.

Kind regards,

Can Yang

Academic Editor

PLOS Computational Biology

Stacey Finley

Section Editor

PLOS Computational Biology

**Additional Editor Comments:**

Two experts in this field have reviewed your manuscript. In general, they are positive on your method, but they still have several major concerns. Please address them in a point-to-point manner in the revision.

**Journal Requirements:**

1) We do not publish any copyright or trademark symbols that usually accompany proprietary names, eg ©,  ®, or TM  (e.g. next to drug or reagent names). Therefore please remove all instances of trademark/copyright symbols throughout the text, including:

- ® on page: 44.

2) Some material included in your submission may be copyrighted. According to PLOSu2019s copyright policy, authors who use figures or other material (e.g., graphics, clipart, maps) from another author or copyright holder must demonstrate or obtain permission to publish this material under the Creative Commons Attribution 4.0 International (CC BY 4.0) License used by PLOS journals. Please closely review the details of PLOSu2019s copyright requirements here: PLOS Licenses and Copyright. If you need to request permissions from a copyright holder, you may use PLOS's Copyright Content Permission form.

Potential Copyright Issues:

i) Figures 1, 2, and 3. Please confirm whether you drew the images / clip-art within the figure panels by hand. If you did not draw the images, please provide (a) a link to the source of the images or icons and their license / terms of use; or (b) written permission from the copyright holder to publish the images or icons under our CC BY 4.0 license. Alternatively, you may replace the images with open source alternatives. See these open source resources you may use to replace images / clip-art:

3) Please amend your detailed Financial Disclosure statement. This is published with the article. It must therefore be completed in full sentences and contain the exact wording you wish to be published.

2) If any authors received a salary from any of your funders, please state which authors and which funders..

4) Please send a completed 'Competing Interests' statement, including any COIs declared by your co-authors. If you have no competing interests to declare, please state "The authors have declared that no competing interests exist". Otherwise please declare all competing interests beginning with the statement "I have read the journal's policy and the authors of this manuscript have the following competing interests"

**Reviewers' comments:**

Reviewer's Responses to Questions

**Comments to the Authors:**

Reviewer #1: Summary

In summary, this paper presents the first double gene knockout (DKO) simulation framework for CRISPR screening in modeling cell growth behavior using mathematical modeling and Monte Carlo simulation. The overall idea is good and the model design is sound. It is very promising and important to extend from single gene knockout (SKO) simulation to double knockout (DKO) modeling.

In general, the introduction and methods are well-written, with clear toy examples that help readers understand the meaning of each parameter and experimental condition. The model itself is also clear and captures the biological process, especially the dynamics of cell growth.

However, in the results section, I have several specific suggestions. Overall, I understand the author’s logic — first demonstrating that the model is tunable, then showing that it can accurately approximate experimental data, and finally confirming that the results are reproducible. Below are my detailed comments for each part.

1. Evaluation of parameter influence on model performance

This section validates the influence of each parameter on model performance using AUC-PR and Pearson correlation. Overall, the evaluation metrics are appropriate, and the results clearly show performance trends with respect to parameter changes. Including the BAGEL method for comparison is also a good choice.

However, I do not think this analysis should appear as the first result — it might fit better later in the results section. Moreover, I suggest evaluating the joint effects of multiple hyperparameters (e.g., changing coverage and another parameter simultaneously) to test whether there are synergistic effects among them.

I also have some questions about the results shown in Fig. 1D–F and Fig. 1J. In Fig. 1D–F, PCC increases monotonically with the proportion of high-efficiency guides, but in Fig. 1J, the PCC is not highest when the rate of high-efficiency guides reaches 100%. I wonder why this occurs — perhaps the authors could perform simulation experiments to analyze the relationship between high-efficiency guide rate and number of guides per gene, rather than only relying on PCC.

Finally, I recommend providing a summary or guideline for hyperparameter selection at the end of this section. There are many tunable parameters (around six), and even though the model is adjustable, users would benefit from practical reference values or examples for parameter tuning.

2. Model approximation to real data

This section evaluates how well the model approximates real experimental data by analyzing data variance (s). However, it is unclear why the authors did not calculate and visualize s in the same way as in Fig. 5F, which would make it easier to directly compare whether the simulation perfectly approximates the experimental data.

The meaning of s (data variance) also needs more explanation — specifically, how it is used to measure approximation quality and why it is a suitable metric. This explanation could be added before the results or in the supplementary materials, with a clear citation in the results part. Otherwise, the role of s is confusing to readers.

Additionally, I think the authors should include a baseline method for comparison. Even though DKOsim may be the first tool for DKO simulation, it would still be informative to slightly modify an existing SKO simulation method and compare it against DKOsim. This would better demonstrate the model’s relative performance and contribution.

3. Reproducibility and biological validation

The third part focuses on the model’s reproducibility and its consistency with real data. However, I find the first part of the experiment somewhat repetitive with Fig. 4 or the first results section. Perhaps the authors could use alternative visualization methods to express these results more clearly.

Since the simulation involves randomness, I also suggest performing the experiments multiple times and adding error bars to show the variability.

The second part of this section presents a good idea, but the validation is currently too limited — relying only on Pearson correlation is insufficient. The authors could further analyze specific examples from both simulated and real datasets, combined with biological background or known gene interactions, to provide stronger validation and insights.

Finally, it would be valuable if the authors could discuss the gene-gene interactions captured in the simulation and compare them with those observed in experimental data, which would highlight the biological relevance of their model.

4. Additional concerns and suggestions

In the end, I still have several concerns about the work.

First, I am not fully convinced by the validation based solely on Pearson correlation (PCC). It seems insufficient to demonstrate that the simulated data are of high quality. I strongly recommend including specific case examples comparing simulated and real datasets, which would make the validation more intuitive and convincing.

Second, although I found a Jupyter notebook on GitHub, there is no clear user documentation explaining how to use this simulation method. The provided analysis_example notebook is quite difficult to follow — the cell_lib section, in particular, is excessively long and affects readability. It might be sufficient to include only a concise example of the cell library instead. In the discussion section, the authors claim that the model is user-friendly, even for users with limited background, but without readable tutorials and proper documentation, this claim is difficult to verify.

Third, I hope to see more insightful biological or practical analyses beyond Pearson correlation, precision, and recall. For instance, it would be interesting to explore whether models trained on this DKO simulated dataset could improve downstream tasks, such as gene expression perturbation prediction, compared with models trained on SKO simulated datasets. Currently, the first set of results feels somewhat preliminary and may be more appropriate as supplementary material.

Finally, I have questions regarding parameter selection, particularly the threshold separating high- and low-efficiency guides at 0.6. Why was 0.6 chosen? Is there any experimental evidence supporting this threshold? Moreover, while the model is tunable, it remains unclear how users should determine suitable parameter combinations in practice. Are these parameters estimated based on real data characteristics or by another optimization strategy? Providing such guidance would greatly improve the model’s usability.

Reviewer #2: Summary:

The authors designed a computational framework to simulate single KO and double KO fitness effects typical for CRISPR screening experiments. Double KO simulation can be used to study and optimize the effects of library parameters in order to obtain optimal gene interaction data from screenings. While the general concept of the paper has its merits and the authors have identified an interesting gap to be addressed, the study itself has weaknesses that can not be ignored: The figures presented in the paper are barely legible due to low resolution and overload of text. The computational frame work is a single R script, that is poorly documented and not according to the standard of today's scientific software development, and the actual usability for other researchers is questionable due to the specific assumptions that the authors make.

Major comments:

- Motivation for the study: the authors state that the two methods to identify genetic interactions (CTG, Gemini) that they used resulted in almost random GI for the original experimental test data, and that this motivated the need for a simulated data set. Why this motivation is understandable, it feels weak because there might be other methods around to calculate genetic interactions, and if not, it should rather motivate the authors to investigate why these methods failed to show good overlap. To use simulated data somehow feels as if the authors try to evade the complexity of real experiments. This leads also to one of the fundamental criticism for this manuscript: how useful is simulated data really, in comparison to real experimental data?

Methods:

"in SKO cells, we assumed a single-guide (sgRNA) disrupts one gene, whereas in DKO screens, two distinct guides were combined as one dual-guide (dgRNA) to perturb two genes simultaneously"

The situation in real experiments is often much more complex: one guide can have quite variable binding, knock-down or knock-out efficiency, which is why real CRISPR library experiments often feature up to 5 guides and more for target a single gene. This makes it questionable how well simulated data can recapitulate actual library experiments.

"knockout of gene 1 will yield, in terms of cell division, one of the following

three outcomes, in one unit of WT cell doubling time: (a) 1 = 0: Cell does not divide and loses viability; (b) 1 = 1: Cell divides once as WT; (c) 1 = 2: Cell divides twice."

A discrete model where cells do divide not at all, equal to WT, or twice as much as WT seems highly unrealistic. From published studies and own work I know that fitness effects are most often subtle changes in growth rate, but still measurable over time, but never 2x WT growth rate. The authors do seem to allow fine-tuning of guide efficacy as stated in Table 1, but it is not clear how this connects with dicsrete model assumption. One foot note reads "3. We are referring to the mean and sd of normal distribution before being truncated to generate the theoretical phenotypes of each class of genes" which points towards transformation of continuous input to the discrete model.

"For every ( 1 ; 2 ) an interaction indicator 1 , 2 was generated. Within the set of DKOs not containing a non-targeting control, we randomly selected % of DKOs and flagged their genes as interacting (coded as = 1) and the rest as not interacting ( = 0)."

There seems to be now way to control which set of genes can be reproducibly flagged as GI-positive or negative, only the percentage. Re-running the same function will then lead to different results every time, due to the random sampling steps? This feels not very reproducible for experimentalists that e.g. want to test small toy data sets, but is probably OK for large data sets.

Results:

- Figure 3 is only text; what is the purpose of this figure? It reiterates the input parameters from table 1 and the mathematical description of the algorithm presented in another chart. This leaves the impression that the authors want to inflate the presented results, which is merely a single R script at the moment.

- Figure 4: not readable at all due to low resolution. Include vector graphics instead of pixel graphics into the manuscript. The line diagrams in the figure show dependence on 1 variable, e.g. coverage, but it is not clear how many simulations were run when changing this parameter, like, was coverage simulated in steps of 10%, 1%, 0.1%, and why is there no variation visible? The curves seem to change abruptly in some of the subfigures (B, E, F, ...), indicating rather sparse simulations? If not please explain the appearance of step-changes, or run more simulations.

- The simulation of library parameters in Figure 4 is actually really interesting and insightful. Only the poor presentation spoils the utility of these results.

- The provided software repository contains mainly a single R script with some documentation added directly to the script. Nowadays it should not be asked too much if software is provided and maintained in a more robust and user-friendly format. Why is it not released as an R package? Building R packages is straight-forward and it could be released on CRAN or Bioconductor (better because more specific to biology). While the math and algorithms seem to be correct, essential elements of scientific code are missing, such as: proper README in mark down; proper LICENSE, packaging as mentioned, with proper version and releases; avoiding hard-coded problematic parameters, e.g. the script uses 100% of all available CPUs without asking (!); modularization, import of functions from sub-scripts; test data and test routines (probably present as additional variants of the script?); user-defined (not hard-coded) output files and paths, ...

- The authors seem to focus on eukaryotic model organisms such as human cell lines. This is OK but limits the usability of this highly specific framework even more. The authors should consider to make parameters like MOI (viral infection) or passages (cell lines) optional, or replace with more general parameters that can be used for bacteria too. Many CRISPR screenings are performed in bacteria (probably more than in human cells), and it feels unnecessary to exclude this audience simply by poor design.

- All in all the manuscript is probably too long and too verbose for the limited scope of the presented results. The authors could try to condense the text and focus on the main findings (related to Figure 4). One example from the results section: "Building upon the tunability of DKOsim, we utilized the designed

systems to approximate the real laboratory Double-CRISPR Knockout screening data,

demonstrating the approachable feasibility of our simulation design in mimicking the CRISPR experiments in real-world settings". This is non-sense, just state the facts in plain English: "We compared the experimental data to the predictions made by our simulation". There are many sections where the results are very verbosely described, which makes it harder (at least in my personal opinion) to extract the main messages.

**Have the authors made all data and (if applicable) computational code underlying the findings in their manuscript fully available?**

The PLOS Data policy requires authors to make all data and code underlying the findings described in their manuscript fully available without restriction, with rare exception (please refer to the Data Availability Statement in the manuscript PDF file). The data and code should be provided as part of the manuscript or its supporting information, or deposited to a public repository. For example, in addition to summary statistics, the data points behind means, medians and variance measures should be available. If there are restrictions on publicly sharing data or code —e.g. participant privacy or use of data from a third party—those must be specified.requires authors to make all data and code underlying the findings described in their manuscript fully available without restriction, with rare exception (please refer to the Data Availability Statement in the manuscript PDF file). The data and code should be provided as part of the manuscript or its supporting information, or deposited to a public repository. For example, in addition to summary statistics, the data points behind means, medians and variance measures should be available. If there are restrictions on publicly sharing data or code —e.g. participant privacy or use of data from a third party—those must be specified.requires authors to make all data and code underlying the findings described in their manuscript fully available without restriction, with rare exception (please refer to the Data Availability Statement in the manuscript PDF file). The data and code should be provided as part of the manuscript or its supporting information, or deposited to a public repository. For example, in addition to summary statistics, the data points behind means, medians and variance measures should be available. If there are restrictions on publicly sharing data or code —e.g. participant privacy or use of data from a third party—those must be specified.requires authors to make all data and code underlying the findings described in their manuscript fully available without restriction, with rare exception (please refer to the Data Availability Statement in the manuscript PDF file). The data and code should be provided as part of the manuscript or its supporting information, or deposited to a public repository. For example, in addition to summary statistics, the data points behind means, medians and variance measures should be available. If there are restrictions on publicly sharing data or code —e.g. participant privacy or use of data from a third party—those must be specified.

Reviewer #1: Yes

Reviewer #2: Yes

PLOS authors have the option to publish the peer review history of their article (what does this mean?). If published, this will include your full peer review and any attached files.). If published, this will include your full peer review and any attached files.). If published, this will include your full peer review and any attached files.). If published, this will include your full peer review and any attached files.

...

Reviewer #1: No

Reviewer #2: **Yes:** Michael JahnMichael JahnMichael JahnMichael Jahn

**Figure resubmission:**
---

## [Decision Letter · Decision Letter 1]

27 Feb 2026

PCOMPBIOL-D-25-01827R1

Double-CRISPR Knockout Simulation (DKOsim): A Monte-Carlo Randomization System to Model Cell Growth Behavior and Infer the Optimal Library Design for Growth-Based Double Knockout Screens

PLOS Computational Biology

Dear Dr. Shen,

Thank you for submitting your manuscript to PLOS Computational Biology. After careful consideration, we feel that it has merit but does not fully meet PLOS Computational Biology's publication criteria as it currently stands. Therefore, we invite you to submit a revised version of the manuscript that addresses the points raised during the review process.

We look forward to receiving your revised manuscript.

Kind regards,

Stacey D. Finley, Ph.D.

Section Editor

PLOS Computational Biology

**Additional Editor Comments:**

The reviewers appreciate the extensive revisions you have completed. Please take the time to address the minor points raised (figures/legends and suggestion to host the vignette publicly for broader impact).

**Journal Requirements:**

**Reviewers' comments:**

Reviewer's Responses to Questions

**Comments to the Authors:**

Reviewer #1: The authors have addressed my comments, especially by reorganizing the tutorial for the simulator.

Moreover, I checked the vignette of DKOsimR and I appreciate the authors for providing it. I suggest that the authors consider hosting the vignette on a website such as ReadTheDocs, which could improve the visibility and impact of the tool.

I also appreciate the authors’ effort to address my concerns, especially the new results focusing on the synergy between the two hyperparameters, which is impressive.

I do not have any further suggestions.

Reviewer #2: The authors have tried to address all of the crucial points, and the manuscript has certainly improved.

I specifically want to thank the authors to bring their computational framework to a higher standard by releasing it as R package.

Without going into details, I would recommend to the authors to once more inspect their figures critically, and remove very small, non-legible text labels, or increase their size. For example, legends in figure 4 for line-type do not allow to distinguish continuous and dashed lines. Also in Figure 4, the color code for coverage changes between D, E, and F, and G, H and I.

Otherwise all important points have been addressed and I do not have further pressing comments.

**Have the authors made all data and (if applicable) computational code underlying the findings in their manuscript fully available?**

The PLOS Data policy requires authors to make all data and code underlying the findings described in their manuscript fully available without restriction, with rare exception (please refer to the Data Availability Statement in the manuscript PDF file). The data and code should be provided as part of the manuscript or its supporting information, or deposited to a public repository. For example, in addition to summary statistics, the data points behind means, medians and variance measures should be available. If there are restrictions on publicly sharing data or code —e.g. participant privacy or use of data from a third party—those must be specified. requires authors to make all data and code underlying the findings described in their manuscript fully available without restriction, with rare exception (please refer to the Data Availability Statement in the manuscript PDF file). The data and code should be provided as part of the manuscript or its supporting information, or deposited to a public repository. For example, in addition to summary statistics, the data points behind means, medians and variance measures should be available. If there are restrictions on publicly sharing data or code —e.g. participant privacy or use of data from a third party—those must be specified. requires authors to make all data and code underlying the findings described in their manuscript fully available without restriction, with rare exception (please refer to the Data Availability Statement in the manuscript PDF file). The data and code should be provided as part of the manuscript or its supporting information, or deposited to a public repository. For example, in addition to summary statistics, the data points behind means, medians and variance measures should be available. If there are restrictions on publicly sharing data or code —e.g. participant privacy or use of data from a third party—those must be specified. requires authors to make all data and code underlying the findings described in their manuscript fully available without restriction, with rare exception (please refer to the Data Availability Statement in the manuscript PDF file). The data and code should be provided as part of the manuscript or its supporting information, or deposited to a public repository. For example, in addition to summary statistics, the data points behind means, medians and variance measures should be available. If there are restrictions on publicly sharing data or code —e.g. participant privacy or use of data from a third party—those must be specified.

Reviewer #1: Yes

Reviewer #2: Yes

PLOS authors have the option to publish the peer review history of their article (what does this mean?). If published, this will include your full peer review and any attached files.). If published, this will include your full peer review and any attached files.). If published, this will include your full peer review and any attached files.). If published, this will include your full peer review and any attached files.

**Do you want your identity to be public for this peer review?** For information about this choice, including consent withdrawal, please see our  For information about this choice, including consent withdrawal, please see our  For information about this choice, including consent withdrawal, please see our  For information about this choice, including consent withdrawal, please see our Privacy Policy....

Reviewer #1: No

Reviewer #2: **Yes:** Michael JahnMichael JahnMichael JahnMichael Jahn

**Figure resubmission:**
---

## [Editor Report · Decision Letter 2]

24 Mar 2026

Dear Dr. Shen,

We are pleased to inform you that your manuscript 'Double-CRISPR Knockout Simulation (DKOsim): A Monte-Carlo Randomization System to Model Cell Growth Behavior and Infer the Optimal Library Design for Growth-Based Double Knockout Screens' has been provisionally accepted for publication in PLOS Computational Biology.

Best regards,

Stacey D. Finley, Ph.D.

Section Editor

PLOS Computational Biology

---

## [Editor Report · Acceptance letter]

PCOMPBIOL-D-25-01827R2

Double-CRISPR Knockout Simulation (DKOsim): A Monte-Carlo Randomization System to Model Cell Growth Behavior and Infer the Optimal Library Design for Growth-Based Double Knockout Screens

Dear Dr Shen,

I am pleased to inform you that your manuscript has been formally accepted for publication in PLOS Computational Biology. Your manuscript is now with our production department and you will be notified of the publication date in due course.

With kind regards,

Judit Kozma
